# Profiling of Saharan dust from the Caribbean to West Africa, Part 1: Layering structures and optical properties from shipborne polarization/Raman lidar observations

Franziska Rittmeister[1], Albert Ansmann[1], Ronny Engelmann[1], Annett Skupin[1], Holger Baars[1], Thomas Kanitz[2], and Stefan Kinne[3]

[1]Leibniz Institute for Tropospheric Research, Leipzig, Germany
[2]ESTEC, Noordwijk, The Netherlands
[3]Max Planck Institute for Meteorology, Hamburg, Germany

*Correspondence to:* A. Ansmann (albert@tropos.de)

**Abstract.** We present final and quality-assured results of multiwavelength polarization/Raman lidar observations of the Saharan air layer (SAL) over the tropical Atlantic. Observations were performed aboard the German research vessel R/V Meteor during the one-month transatlantic cruise from Guadeloupe to Cabo Verde over 4500 km from 61.5°W to 20°W at 14-15°N in April-May 2013. First results of the shipborne lidar measurements, conducted in the framework of SALTRACE (Saharan Aerosol Long-range Transport and Aerosol-Cloud Interaction Experiment), were reported **by Kanitz et al. (2014)**. **Here we present** four observational cases representing key stages of the SAL evolution between Africa and the Caribbean in detail in terms of layering structures and optical properties of the mixture of predominantly dust and aged smoke in the SAL. We discuss to what **extent** the lidar results confirm the validity of the SAL conceptual model which describes the dust long-range transport and removal processes over the tropical Atlantic. Our observations of a clean marine aerosol layer (MAL, layer from the surface to SAL base) confirm the conceptual model and suggest that the removal of dust from the MAL, below the SAL, is very efficient. However, the removal of dust from the SAL assumed in the conceptual model to be caused by gravitational settling in combination with large scale subsidence is weaker than expected. To explain the observed homogenous (height-independent) dust optical properties from SAL base to SAL top, from the African coast to the Caribbean, we have to assume that the particle sedimentation strength is reduced and dust vertical mixing and upward transport mechanisms must be active in the SAL. Based on lidar observations in 20 nights at different longitudes in May 2013, we found, on average, MAL and SAL layer mean values (at 532 nm) of the extinction-to-backscatter ratio (lidar ratio) of 17±5 sr (MAL) and 43±8 sr (SAL), of the particle linear depolarization ratio of 0.025±0.015 (MAL) and 0.19±0.09 (SAL), and of the particle extinction coefficient of 67±45 Mm$^{-1}$ (MAL) and 68±37 Mm$^{-1}$ (SAL). The 532 nm optical depth of the lofted SAL was found to be, on average, 0.15±0.13 during the ship cruise. The comparably low values of the SAL mean lidar ratio and depolarization ratio (compared to typical pure dust values of 50–60 sr and 0.3, respectively) **in combination with backward trajectories** indicate a smoke contribution to light extinction of the order of 20% during May 2013, at the end of the burning season in central-western Africa.

# 1  Introduction

Dust particles can travel over long distances of more than 10000 km in the free troposphere (Haarig et al., 2017a) and can be lifted up to the tropopause under favorable meteorological conditions during long-range transport (Mamouri and Ansmann, 2015; Hofer et al., 2016, 2017). Dust is an important component of the northern hemispheric aerosol system and sensi-
tively influences environmental and climatic conditions on the regional to intercontinental scale (Myhre and Stordal, 2001; Sokolik et al., 2001; Tegen, 2003; Balkanski et al., 2007). The impact of mineral dust on the evolution and lifetime of liquid-water, mixed-phase, and ice clouds via heterogenous ice formation is presently in the focus of atmospheric research (Seifert et al., 2010; Murray et al., 2012; Hoose and Möhler, 2012; Atkinson et al., 2013; Phillips et al., 2013; DeMott et al., 2015; Kiselev et al., 2017).

The Saharan desert is the world's largest mineral dust source (Prospero et al., 2002; Cakmur et. al., 2006; Huneeus et al., 2011) and the transport of mineral dust across the tropical Atlantic is the most prominent example of a powerful long-distance transport of mineral dust. Karyampudi et al. (1999) presented a conceptual model to describe the transport and deposition of dust during advection from Africa to America. The model originated from the work of Carlson and Prospero (1972). We introduce this conceptual model in Sect. 1.1. Dust advection across the tropical Atlantic during the summer months is almost
not affected by anthropogenic pollution so that changes in the dust characteristics during transport can be studied in large detail. The pioneering work of J. Prospero (Prospero, 1968; Prospero et al., 1972), which he began more than 50 years ago at Barbados in August 1965, triggered numerous dust research activities and well organized field campaigns in western Africa and over the tropical Atlantic. Overviews of these advanced field campaigns conducted during the last 15 years are given in Ansmann et al. (2011) and Ryder et al. (2015).

To investigate dust and its climate-relevant aspects comprehensive dust field experiments (ground-based and airborne activities, in situ measurements combined with active and passive remote sensing) are required with focus on the complex relationship between the microphysical, chemical, morphological shape, optical, radiative, and cloud-process-relevant properties of dust particles. The latest attempt to characterize dust over scales of several thousands of kilometers of travel distance (equivalent to 5-10 days of travel time) has been performed by the series of well-defined field activities: The Saharan Mineral Dust
Experiments SAMUM-1 (southern Morocco, summer 2006) (Heintzenberg, 2009) and SAMUM-2 (Cabo Verde, in the winter and summer of 2008) (Ansmann et al., 2011), and the Saharan Aerosol Long-Range Transport and Aerosol-Cloud-Interaction Experiment SALTRACE (Barbados, in the summers of 2013 and 2014, and the winter of 2014) (Weinzierl et al., 2017).

Winter as well as summer-mode dust transport regimes (Schepanski et al., 2009; Ben-Ami et al., 2009) were covered by the two SAMUM-2 and three SALTRACE field phases (Tesche et al., 2011a; Haarig et al., 2017a). Based on the SAMUM-2
observations and simultaneously performed lidar measurements in Amazonia it was clearly demonstrated for the first time that not only desert dust, but also significant amounts of biomass burning smoke are transported towards South America (Ansmann et al., 2009; Tesche et al., 2011b; Baars et al., 2011) and even sporadically to the Caribbean during the winter half year (Haarig et al., 2016). The dust/smoke layers are advected at comparably low heights, typically below 3 km height, during the winter season. In contrast, almost pure dust plumes leave the African continent in summer. The dust layers reach up to

5-6 km height over western Africa (Tesche et al., 2011a) and the eastern part of the tropical Atlantic in summer. The dust is then found over Barbados between about 1.5 km and 4 km height (Groß et al., 2015; Haarig et al., 2016, 2017a).

The main goal of the SAMUM and SALTRACE activities was to conduct a detailed vertically resolved characterization of Saharan dust close to the source as well as within the Saharan air layer (SAL) on the way towards the Caribbean, more than 5000 km downwind the main source regions. Well designed efforts of combined airborne and ground-based in situ aerosol observations and remote sensing were realized during all of the three campaigns. To better link the SAMUM and SALTRACE results, continuous lidar observations were conducted aboard the German research vessel R/V Meteor during a cruise from Guadeloupe to Cabo Verde from 29 April to 23 May 2013, and thus during the transition period from winter to summer dust transport conditions. The SAMUM and SALTRACE field sites (Morocco, Cabo Verde, Barbados) and the R/V Meteor cruise are shown in Fig. 1 (top).

The fully automated multiwavelength polarization/Raman lidar (Engelmann et al., 2016) aboard the research vessel measured height profiles of optical properties of the particles in the marine aerosol layer (MAL, dominated by sea salt aerosol) and of the mixture of predominantly dust and smoke in the SAL. Over the remote tropical Atlantic, the MAL comprises the convection marine boundary (MBL) and a frequently occurring residual, less convective layer. The MAL reaches up to the trade wind inversion height (height of a strong temperature inversion, usually between 1500 and 2500 m height) (Augstein et al., 1973; Cao et. al., 2007) over the undisturbed open Ocean. Kanitz et al. (2014) provided a first overview of the shipborne lidar observations across the Atlantic, discussed the general features of dust layering during the cruise, and compared the dust profile structures with preliminary dust modeling results. The authors also presented two contrasting cases (cases 1 and 4 in Fig. 1 for very fresh and aged dust plumes, respectively) in terms of vertical profiles of particle backscatter and extinction coefficients, extinction-to-backscatter or lidar ratio, linear depolarization ratio, and Ångström exponents (describing the wavelength dependence of backscatter and extinction).

In a series of two articles, we now present the final results of the cruise. In part 1, we discuss the dust layering characteristics in large detail (Sect. 3.2) and the aerosol optical properties of four cases (Sect. 3.3), which we denote as key stages of the SAL evolution across the Atlantic. **We discuss the intermediate stages of dust transport in addition to the two stages of dust layering shortly after emission and about 10 days after long-range transport across the Atlantic. These two cases with fresh and aged dust were already presented by Kanitz et al. (2014).** We compare our R/V Meteor lidar observations with the main features of dust transport and removal over the Atlantic Ocean as described by the conceptual model developed by Karyampudi et al. (1999) in Sects. 3.2.1 and 3.3.1. The conceptual model is explained in Sect. 1.1. We discuss the mixing state of dust and non-dust aerosol components in the SAL in Sect. 3.3. All available cloud-free nighttime Raman lidar observations (20 nights within the 1–23 May 2013 period) were analyzed and the vertical mean values of particle extinction, lidar ratio, and depolarization ratio for MAL and the lofted SAL were determined. These results are presented in Sect. 3.3 as well. In part 2 (Ansmann et al., 2017), the selected four key observational cases are further analyzed by applying the Polarization Lidar Photometer Networking (POLIPHON) method of Mamouri and Ansmann (2017). Fine dust, coarse dust, and non-dust backscatter contributions are separated and dust, marine, and smoke/haze contributions to light extinction and particle mass concentration are quantified in part 2 and compared with respective dust forecasts of three well-established dust models.

## 1.1 Conceptual model

Large quantities of Saharan dust are transported across the tropical Atlantic throughout the year but more abundantly in summer months. These dust outbreaks are mostly confined to a deep mixed layer, commonly referred to as the Saharan air layer (SAL) that often extends to 5-6 km in height over West Africa due to intense solar heating in summer months. The conceptual model (Karyampudi et al., 1999) describes the dust transport from Africa to North America in detail. According to this model, which is based on research performed in the 1970s to 1990s, the hot, dry, and dust-laden air masses emerge from the western coast of Africa as a series of large-scale pulses in the summer months. Associated with easterly wave activity, Saharan dust outbreaks occur predominantly within the ridge region of passing easterly wave disturbances and generally last 3–5 days. The airborne dust is carried westward by the prevailing easterly flow in the latitude belt of 10°–25°N. As the dust plumes approach the West African coastline and are advected further west in the easterly flow, the base of the SAL rises rapidly as it is undercut by the relatively clean northeasterly trade winds, which are confined within the MBL, while the **SAL** top subsides slowly. Because of the low-level aerosol erosion by the clean trade winds, the major transport of dust occurs in an elevated well-mixed layer (i.e., within the SAL).

As we will show in Sect. 3, the well-mixed SAL resides above a relatively dust-free layer, which we denote as marine aerosol layer (MAL). As mentioned, the MAL comprises the humid, convective MBL, the top of which is frequently capped by cumulus clouds, and the occasionally occurring residual, less convective layer from MBL top to SAL base. This residual layer below the SAL (or, if absent, the upper part of the MBL) is denoted as trade wind inversion layer in the conceptual model. Marine aerosol particles dominate the backscattering and extinction properties in the MAL according to our shipborne lidar measurements. Note that another way to divide the vertical column up to SAL base (or the trade wind inversion) is related to cloud formation and occurrence. The layer up to the base of trade wind cumuli is called sub-cloud layer. The remaining layer from cloud base to SAL base or trade wind inversion is denoted as cloud layer (e.g., Siebert et al., 2013).

The dust transport takes usually 5-7 days across the Atlantic. While the SAL base rises with distance from Africa (from about 1500 m (850 hPa) height in the Cabo Verde region to about 2500 m (750 hPa) in the Barbados region), the SAL top gradually lowers from 5–6 km (Cabo Verde) height to 4–5.5 km height (Barbados), most likely induced by subsidence associated with the Hadley circulation. Karyampudi et al. (1999) suspected that the comparably rapid sinking of the dust layer top over the eastern Atlantic may be due to the rapid depletion of giant particles away from the west African coast line.

The strong temperature inversion at the base of the SAL limits convective activity within the SAL and consequently precludes the possibility of strong wet removal from the dust layer (scavenging of dust particles below and within clouds and removal by wash and rain out), except during periods with deep convection and precipitation. Karyampudi et al. (1999) conclude that the rapid decrease of the high dust concentration over the eastern Atlantic is most probably related to settling of heavy dust particles from the top to lower heights as the plume progresses westward. The sub layers (below the SAL) receive dust particles by vertical downward mixing and mainly by gravitational setting (fallout) from the overlaying SAL. Convective mixing is pushing clean MBL air up in altitude and mass conservation forces dusty air to low altitudes, where it is efficiently removed by scavenging within and below clouds (rain out) or turbulent downward mixing. The residual aerosol layer located between

the top of the convectively active MBL and the SAL possibly represents a mixture of mineral aerosol from the SAL above and sea salt aerosol from the MBL below. As the dust is advected west the low-lying material is eroded away by wet removal or dry deposition.

According to the modeling study of Colarco et. al. (2003) (with detailed focus on dust removal), the vertical distribution of dust over the tropical North Atlantic appears to be most strongly controlled by particle sedimentation and the general descent of air (downward vertical velocity) during the transit between Africa and Central America and north of the ITCZ. Wet removal does not greatly affect the vertical distribution of the dust except through removal of dust within the MBL. Some of the low-lying dust may persist all the way across the ocean. Turbulent mixing caused by wind shear in the height range around the trade wind inversion zone (MAL upper part, SAL lower part) may contribute to the removal of dust from the SAL and entrainment into the upper MAL. Convective mixing and wet removal in the MAL efficiently transport the dust down to low heights.

The question now arises: are these features of the dust transport across the Atlantic as described by the conceptual model in agreement with our shipborne lidar observations? Are there aspects that are not described and/or considered properly but have an impact on dust transport and removal? The discussion is presented in Sects. 3.2 and 3.3. Before we present the results in Sect. 3 , we briefly describe the R/V Meteor cruise, the lidar and other atmospheric measurement instruments used aboard the ship, and the basic lidar retrieval methods.

## 2 SALTRACE R/V Meteor cruise and instrumentation

The first vertically resolved lidar-based study of the SAL across the tropical Atlantic was presented by Karyampudi et al. (1999) based on the space lidar LITE (Lidar In-Space Technology) observation aboard the Space Shuttle Discovery in September 1994 (McCormick et al., 1993). Systematic studies of the east-to-west dust transport with the satellite lidar CALIOP (Cloud and Aerosol Lidar with Orthogonal Polarization) were then presented by Liu et al. (2008a); Liu et al. (2008b). Further Saharan dust studies over the tropical Atlantic based on CALIOP measurements can be found in Adams et al. (2012) and Tsamalis et al. (2013). The latter authors characterized the decay of the Saharan dust amount in terms of layer descent and deposition velocity.

However, all of these lidar studies are based on observations with standard elastic-backscatter lidars which do not have the potential to directly measure the particle extinction coefficient in the SAL and thus of the SAL aerosol optical thickness (AOT). One of the most important input profile in the retrieval of the basic lidar products with these standard lidars is the height profile of the particle extinction-to-backscatter coefficient (lidar ratio). This input profiles has to be estimated and can be very erroneous. In the case of LITE, the dust lidar ratio was assumed to be in the range from 20-35 sr for pure dust (assuming spherical dust particles). Since 2002 observed Saharan dust lidar ratios are available (Mattis et al., 2002). The Raman lidar observations during the SAMUM campaigns then corroborated that the desert dust lidar ratio for western Saharan dust is between 45-60 sr (Wandinger et al., 2010; Tesche et al., 2011a, 2013) and also that the lidar ratio may vary with height as a function of dust mixture with marine and smoke particles as we will show below. Compared to the LITE and CALIOP studies advanced aerosol lidars, such as the multiwavelentgh polarization/Raman lidar as applied her, provide a wealth of aerosol information on the aerosol mixture and dust optical and microphysical properties, including the profile of the dust extinction

coefficient, the SAL AOT, and the profile of the lidar ratio, so that we have a much better basis for profiling and characterizing the aerosol in the SAL and to study the changes in the aerosol properties (aging of the aerosol) during long range transport. Part 1 (this article) and part 2 (Ansmann et al., 2017) show the full potential of modern aerosol lidars.

## 2.1 R/V Meteor cruise

The transatlantic cruise M96 of the German R/V Meteor took place from 29 April to 23 May 2013 starting at Guadeloupe (16° N, 61° W) and ending at Cape Verde (17° N, 25° W). The journey covered a distance of approximately 4500 km (Fig. 1, top). The containerized OCEANET-Atmosphere platform (Kanitz et al., 2011, 2013) aboard is usually operated during north-south cruises of R/V Polarstern between Bremerhaven, Germany, and Cape Town, South Africa, or Punta Arenas, Chile.

## 2.2 Polly$^{XT}$

The multiwavelength Raman/polarization lidar system Polly$^{XT}$ (Engelmann et al., 2016) is the key instrument of the OCEANET-Atmosphere platform and installed inside the container. Polly stands for *POrtabLle Lidar sYstem*, XT for extend version. The lidar performed continuous observations during the four-week travel. By means of a two-telescope receiver arrangement for near-range and far-range tropospheric profiling, aerosol extinction profiles (computed from smoothed Raman signal profiles) are available from 400 m height up to cirrus and tropopause level and thus cover the upper part of MBL, most of the MAL

and the entire SAL vertical range. Full overlap of the near–range telescope receiver field-of-view with the laser beam is at about 200-250 m. However, the full set of lidar products allow us to study the vertical distribution of dust and marine aerosol particles down to 100 m above sea level, because the determination of profiles of the backscatter coefficients and the linear depolarization ratio is based on the analysis of profiles of signal ratios and these profiles are available with good accuracy down to very low heights because the overlap effect widely cancels out. The advanced aerosol lidar enabled us to measure profiles

of the particle backscatter coefficients (180° scattering coefficient) at 355, 532, and 1064 nm, particle extinction coefficient at 355 and 532 nm, the respective lidar ratios (extinction-to-backscatter ratios) as well as the particle linear depolarization ratio at 355 and 532 nm (Freudenthaler et al., 2009; Engelmann et al., 2016; Baars et al., 2016). A discussion of the uncertainties in the retrieval products is also given in these articles.

  Kanitz et al. (2014) presented first results based on a given set of available input parameters in the lidar data analysis. After

careful re-analysis of all input parameters, we now present the final results and summarize the main findings. Most important input parameters are the overlap profiles between the laser beam and the receiver field-of-view for the near-range and the far-range telescopes and the parameters describing the performance of the lidar transmitter and receiver optics, and thus the impact of the receiver optical properties on the determination of the linear depolarization ratios (see Haarig et al., 2017 for details of this procedure in the case of our SALTRACE Barbados lidar). **The final quantitative data values** presented here differ only

slightly from the ones of (Kanitz et al., 2014), except for the profiles obtained with the near-range telescope and the profiles of the 355 nm particle linear depolarization ratio.

  One of the most important lidar-derived aerosol parameter in dust observations is the volume depolarization ratio. This quantity is almost directly measured. The volume linear depolarization ratio is obtained from the calibrated ratio of the cross-

to-co-polarized backscatter signal (Freudenthaler et al., 2009). Co and cross denote the planes of polarization (for which the receiver channels are sensitive) parallel and orthogonal to the plane of linear polarization of the transmitted laser pulses, respectively. The volume depolarization ratio is influenced by light depolarization by air molecules and aerosol and cloud particles. To obtain the particle depolarization ratio a correction for Rayleigh depolarization effects has to be applied (Freudenthaler et al., 2009).

The particle depolarization ratio allows us to separate fine dust (dust particles with diameters $<1$ $\mu$m), coarse dust (super micron particles), and the non-dust aerosol components by applying the POLIPHON technique (Mamouri and Ansmann, 2014, 2017). This aerosol separation technique is based on characteristic particle linear depolarization ratios for spherical marine particles, 0.02–0.03, for urban haze and biomass burning smoke, $\leq 0.05$, and around 0.3 for desert dust at 532 nm (Sugimoto et al., 2003; Shimizu et al., 2004; Tesche et al., 2009). Important assumptions are also that dust and non-dust particles are externally mixed and that the dust particles are hydrophobic so that water uptake at high humidities is negligible. The corresponding analysis of the R/V Meteor lidar observations is given in the follow-up article (Ansmann et al., 2017).

Part of the results on the aerosol mixing state in the SAL are already discussed here (in part 1) so that the following sources of uncertainty need to be mentioned. In discussions, frequently the question arises to what extend dust aging by chemical processes, activation of dust particles to serve as cloud condensation nuclei in cloud formation events, and by size-dependent gravitational settling during long-range transport affect the accuracy of the separation of dust from non-dust aerosol profiles. The assumption of a universal dust depolarization ratio of about 0.3 may not be generally valid. Aging of dust particles caused by cloud evolution and subsequent evaporation processes or by chemical reactions on the dust particle surface (Abdelkader et al., 2015, 2017) may change the chemical composition and shape characteristics (towards more smoothed, more spherical, less irregular shaped dust particles) and thus may lead to a significant decrease of the dust depolarization ratio during long-range transport. Furthermore, a stronger removal of large dust particles compared to small particles by gravitational settling may change the dust particle size distribution significantly. A shift of the particle size distribution towards smaller particles should be reflected in a decreasing particle depolarization ratio. Fine dust causes particle linear depolarization ratios of 0.14–0.18 at 532 nm, whereas the dust coarse mode leads to values between 0.35–0.4 (see discussion in Mamouri and Ansmann, 2017). However, the literature published on dust optical properties does not support a strong dust aging effect. Weinzierl et al. (2017) showed dust size distributions measured over Cabo Verde in June 2013 and over Barbados 4-5 days later. The observed removal of the large dust particles, predominantly of particles with diameters $>5$ $\mu$m, only caused a minor reduction in the dust depolarization ratio from about 0.31 (SAMUM-2, Cabo Verde) to about 0.28 (SALTRACE, Barbados) at 532 nm (Haarig et al., 2017a). Changes in the dust shape characteristics by cloud processes and chemical aging obviously have no impact on the dust depolarization ratio.

Internal mixing of dust and non-dust particles may cause a misinterpretation of the dust and non-dust particle extinction coefficients and mass concentrations retrieved with the POLIPHON method. The internally-mixed non-dust aerosol contributions will be counted as dust contributions. Chemical aging and successive water uptake at high humidity may significantly change the overall (dust and non-dust) particle extinction coefficient **(Abdelkader et al., 2017)** and thus the retrieval of dust mass concentrations from the lidar-derived particle extinction coefficients.

Lieke et al. (2011), Kandler et al. (2011), Denjean et al. (2015), and Weinzierl et al. (2017) investigated to potential impact of cloud processing and chemical aging and analyzed their observations regarding a potential sulfate coating of dust particles or a potential internal mixing with marine (sea salt) aerosol components. According to the airborne in-situ measurements performed during the SALTRACE over Cabo Verde and in the Barbados region in the summer of 2013 (Weinzierl et al., 2017), the abundance of silicate particles (in the diameter range from 500–1000 nm) with detectable amounts of sulfate ($\approx$ 1%) increased from 2.5% (in the Cabo Verde region) to 4.3%(in the Caribbean). For supermicron particles, no considerable modification (internal mixture) was visible. Denjean et al. (2015) performed detail chemical analysis of long-range transported dust with diameters >1 $\mu$m in Puerto Rico (in situ observations at ground) in the summer of 2012 and found that about 10% of the dust particles were internally mixed and showed sulfate and chloride coatings during a strong dust event and 3–6% of the dust particles showed an internal mixture with sea-salt particles. Lieke et al. (2011) (airborne observations) and Kandler et al. (2011) (ground-based observations) found during the SAMUM-2 campaign (Cabo Verde, January-February 2008, biomass burning season in central-western Africa) internal silicate-sulfate mixtures in roughly 10-20% of the analyzed particles in the 500–1000 nm range, and about 10% for particles with diameters >1 $\mu$m. An internal mixture of dust and soot was not observed in the free tropospheric layers (Lieke et al., 2011).

Regarding potential changes of the particle optical properties, Denjean et al. (2015) observed that internal mixing did not have any effect on the hydrophobic behavior of dust, the optical properties did not change for low and high relative humidities. Groß et al. (2016) found a good correlation between in-situ observed dust mass concentrations at Ragged Point, Barbados, and lidar-derived dust mass concentrations within the humid MBL with typical relative humidities around 80%. This corroborates that the assumption of an external mixture of hydrophobic dust and non-dust aerosol particles is sufficiently valid, and also that the used assumption of a dust depolarization ratio of 0.3 at 532 nm in the lidar retrieval is valid not only close to the Sahara but also after long-range transport over 5000 km. If we finally assume that the mass concentration of, e.g., sulfate to the total (silcate + sulfate) particle mass concentration of internally mixed particles (10% of all particles) is < 10% than the effect of internal mixing on the POLIPHON results (dust extinction coefficient and mass concentration) is of the order of <1% and thus negligible.

## 2.3 Sun photometer

The lidar profile observations were accompanied by sun photometer measurements in the framework of the Maritime Aerosol Network (MAN) as part of the Aerosol Robotic Network (AERONET) (Smirnov et al., 2009). The MICROTOPS II measurements provide column-integrated aerosol optical properties at 440, 500, 675, 870, and 936 nm (MAN, 2016). In this work, we will use the 440-870 nm Ångström exponent and 500 nm AOT. AOT uncertainties are around $\pm$0.02 for each AOT channel.

## 2.4 Auxiliary observations and data

Radiosondes for measuring temperature, humidity, and wind profiles were regularly launched at noon and midnight UTC by the German Weather Service on board the ship (AWI, 2016). The temperature and pressure profiles are used to compute the Rayleigh backscattering and extinction contributions to the observed lidar return signals. Missing radiosonde information

has been filled with GDAS (Global Data Assimilation System) height profiles of temperature and pressure from the National Weather Service's National Centers for Environmental Prediction (NCEP) (GDAS, 2016). Aerosol sources apportionment analysis has been supported by air mass transport computation with the NOAA (National Oceanic and Atmospheric Administration) HYSPLIT (HYbrid Single-Particle Lagrangian Integrated Trajectory) model (HYSPLIT, 2016) using GDAS meteorological data (Stein et al., 2015). The backward trajectories have been used in conjunction with maps of active fires as determined with MODIS (Moderate Resolution Imaging Spectroradiometer) on board the Terra and Aqua satellites (MODIS, 2016).

## 3 Results

### 3.1 R/V Meteor cruise overview

Figure. 1 (center panel) provides an overview of dust layering during the R/V Meteor cruise between 2 May and 23 May 2013. The figure is taken from Kanitz et al. (2014). The Saharan air layer, SAL, reached from the top of the marine aerosol layer, MAL, to about 3–5 km height. The MAL top height is indicated by a white line in Fig. 1. In this marine layer, sea salt particles are mainly responsible for the measured optical properties, dust was almost absent. The particle depolarization ratio was low with values around 0.05 or less. The MAL was very shallow (partly only 250–500 m thick) between Cabo Verde and the West African coast (20-23 May 2013) and typically between 1000–1500 m high in the remote marine atmosphere west of Cabo Verde. In the SAL, the particle depolarization ratio was usually >0.15 and indicated the predominance of dust particles.

Figure 1 (bottom panel), also taken from Kanitz et al. (2014), provides an overview of the aerosol optical thickness (AOT) at 500 nm wavelength during the cruise. The daily mean 500 nm AOT ranged from about 0.05, which is indicative here for clean marine aerosol conditions, to 0.7 during the major dust outbreak, observed at Mindeloh, Cabo Verde, at the the end of the cruise (case 1). During times with lofted dust (gray-shaded areas in Fig. 1, bottom) the AOT was mostly between 0.1–0.3. In addition, the Ångström exponent calculated from sun photometer AOT measurements in the 440–870 nm spectral range is shown in Fig. 1 (bottom). The Ångström exponent for marine particles is typically between 0.3–0.7 and drops towards very low values (close to zero) when desert dust with a strong coarse-mode fraction is present.

Three phases of dust transport are visible in Fig.1 (center and bottom panels). Close to Africa, SAL base height is low and not well defined. The column Ångström exponent is very low and indicates the strong impact of dust on the aerosol optical properties. Then the MAL top height or SAL base height stabilizes with values from 0.7–1.7 km height (downwind, towards the Caribbean). The Ångström exponent increases to values around 0.3, which suggests the still strong impact of lofted dust on the observed optical properties. Finally, in phase 3, the Ångström exponent increases to values around 0.5-0.6 which is indicative for the increasing influence of finer aerosol on the column optical properties. The dust impact decreases significantly for sites west of 45°W.

### 3.2 Case studies of aerosol layering

We selected four cases to study the evolution of the SAL and changes in the dust optical properties with increasing distance from Africa in detail. They are indicated by numbers 1–4 in Fig. 1. Figure 2 shows these four lidar observations representing key stages of aerosol layering over the tropical Atlantic. Case 1 and case 4 were already shown by Kanitz et al. (2014). They provide insight into the optical properties of a fresh and very aged dust plume, respectively. According to the HYSPLIT backward trajectories, discussed in Sect. 3.3, the aged dust plume observed on 5 May 2013 (case 4) traveled 9 days across the Atlantic before reaching the research vessel at 53°W. In contrast, the fresh dust plumes (case 1) crossed the shipborne lidar at Mindeloh, Cabo Verde, after approximately 1 day over the ocean. Cases 2 and 3 were then selected to have three cases (1, 2, and 3) with linearly increasing temporal distance from the African west coast (1, 3, and 5 days).

The most remarkable features in Fig. 2 is the sharp increase of the volume depolarization ratio at the base height of the SAL from <0.03 in the MAL (marine particles dominate the volume depolarization ratio and cause very low light depolarization) to >0.1 in the SAL where the strongly light-depolarizing dust particles dominate (see cases 2 and 3). Note also the steady increase of the vertical extent of the MAL with distance from Africa (cases 1–3). Another noticeable finding is the decrease of the SAL depth with distance from Africa. Both findings (increase of MAL depth, decrease of SAL depth) confirm the conceptual model (see Sect. 1.1).

To better understand the observed aerosol layering structures and vertical exchange processes over the tropical ocean and to check the consistency with the conceptual model in Sect. 3.2.1, we introduced the three layers MBL, MAL, and SAL. The top of the MBL is determined from the range-corrected signal profiles (shown in the left panel) by using the gradient method (Baars et al., 2008). At the top of the convective and humid MBL, the range-corrected lidar signal usually drops significantly even during cloud-free conditions. The layers showing a low depolarization below the SAL in the right panels of Fig. 2 indicate the MAL. Under undisturbed marine conditions west of Cabo Verde the MAL indicates the trade wind zone capped by a strong temperature inversion. The top of the MAL is defined by the strong increase of the volume depolarization ratio at the interface between MAL and SAL (e.g., at about 1 km height in case 2) and can thus easily be obtained from the lidar observations. The vertical extent of the MAL does not necessarily be equal to the depth of the MBL. This was the case on 9 May 2013 (case 3), when the MAL was much deeper than the MBL. In Fig. 2 (cases 2-4), the white spots in the left panels indicate trade wind cumuli which developed in the upper part of the MBL.

As was often observed during fair weather conditions (during SALTRACE over Barbados, during SAMUM-2 over Cabo Verde), cumulus convection can intensify and then the vertical extent of the clouds can increase from 300-500 m to 1-2 km. At these conditions, stratocumulus fields tend to develop at the top of the cloud active zone (below the trade wind inversion). The clouds evaporate later on and leave behind a marine aerosol up to SAL base (or MAL top). Such a situation may have led to the aerosol layering as observed on 9 May 2013 (case 3 in Fig. 2). A vertically deep, cloudless MAL (trade wind zone) is detected reaching to almost 2 km height, while the depth of the MBL was <1 km as the white cumulus cloud spots in the right panel of Fig. 2 for case 3 indicate. The upper part of the MAL, from MBL top to MAL top, may be interpreted as a residual marine boundary layer which developed during times (hours to days before the lidar observation) with strong cumulus convection and

vertical mixing (upwind the R/V Meteor). This residual aerosol layer is also mentioned in the conceptual model in Sect. 1.1 when discussing the undercutting effect occurring below the trade wind inversion height. However, one should emphasis that this is a hypothesis. We have no possibility to **prove this argument.**

### 3.2.1  Conceptual model versus lidar observations (part 1)

To facilitate the comparison between the lidar observations in Fig. 2 and the dust layering features and deposition aspects as described in the conceptual model in Sect. 1.1, we show a sketch (Fig. 3) of the observed dust layering in Fig. 2. The sketch highlights the different layers MBL, MAL, and the SAL and indicates the main vertical exchange mechanisms in the conceptual model such as gravitational settling of dust in the SAL and turbulent, convection and wind-shear-induced downward mixing of dust in the layers below the SAL. The top panel illustrates the undercutting effect, i.e., advection of clean air from the

Northeast, below the SAL base (the height of the trade wind inversion over the open Ocean, cases 2–4). In the SAL, the dust is transported from easterly directions. This is partly well confirmed by the snapshot-like shipborne radiosonde profiles. For example, in the evening of 14 May 2013 (case 2), wind speed was about 10 m/s in the MAL and SAL, increased up to 14 m/s in the lower part of the SAL (lowest 500 m), and the wind direction was about $60°$ in the MAL and 100-120$°$ in the SAL.

Wet deposition during times with clouds and precipitation as well as dry deposition contribute to the dust removal from the

MAL. We analyzed METEOSAT satellite observations for the presence of strong cumulus convection and found that, except for case 4, wet deposition by deep convection and associated rain can be excluded. However, fair weather cumulus convection and light precipitation always occurs over the tropical Atlantic and thus a certain contribution of wet deposition to dust removal must be always taken into account.

Our observations are to a large extent in good agreement with the conceptual model. As already mentioned in the foregoing

subsection, the observed changes in the MAL top height (increase), SAL base height and top height (decrease) and SAL vertical extent (decrease) with distance from Africa are similar to the ones described in the conceptual model. The observed sharp increase of the volume depolarization ratio at the interface between MAL and SAL suggests that injection of particles from the MAL into the SAL over the open Atlantic is almost impossible. Furthermore, the rather low depolarization ratio from the ocean surface to MAL top suggests a fast and efficient removal of dust from the MAL by the vertical exchange processes.

It is noteworthy to mention that less sharp, more smooth structures in the depolarization ratio at the MAL/SAL interface were observed over the western part of Barbados during the SALTRACE summer campaigns in June and July 2013 and 2014 (Groß et al., 2016; Haarig et al., 2017a). Heat island effects associated with an enhanced turbulent air flow and stronger downward mixing over Barbados was probably responsible for the observed strong downward transport of dust from the SAL into the MAL (Engelmann et al., 2011; Jähn et al., 2016; Chouza et. al., 2016a).

Gravitational settling plays the dominant role in the downward transport of dust in the SAL over the open Atlantic according to the conceptual model. However, our detailed profile observations presented in Sect. 3.3 as well as the SALTRACE observations at Barbados in June and July 2013 and 2014 (Groß et al., 2015; Gasteiger et al., 2017; Haarig et al., 2017a) suggest that further processes in the SAL are active in addition and counteract sedimentation of dust particles. We will continue the discussion of this point in Sect. 3.3.1, after the introduction and detailed explanation of the optical properties in the next section.

## 3.3 SAL and MAL optical properties

In Fig. 4, the vertical profiles of the derived optical properties for cases 1–4 are presented. The basic lidar signals were averaged (over 20–75 minutes) and vertically smoothed with window lengths of 457 and 563 m to reduce the uncertainty in the products caused by signal noise. Therefore, sharp changes in the profile as visible in Fig. 2 at SAL base obtained with temporal and vertical resolution of 30 s and 7.5 m are considerably smoothed out in Fig. 4.

A strong dust outbreak was observed on 23 May at Cabo Verde (case 1). According to the backward trajectories in Fig. 5 the dust layers arrived at Mindeloh, Cabo Verde, after a short travel over the tropical Atlantic and accumulated dust over 3–4 days. The impact of biomass burning smoke was low in this case. The analysis of the lidar observation regarding the contribution of non-dust aerosol components such as marine particles, urban haze, and fire smoke particles to the total particle backscatter and extinction profiles is given in part 2 (Ansmann et al., 2017) and shows that the non-dust contribution to light extinction at 532 nm was of the order of 10%. As outlined in Sect. 2.2, most of the non-dust aerosol was probably externally mixed. The daily mean value of the AERONET Ångström exponent in Fig. 1 is rather low (0.1) and indicates the dominance of large particles in the lofted aerosol layer.

The backward trajectories for case 2 (15 May 2013, 00:00 UTC) suggest a possible impact of smoke in the upper half of the SAL, i.e., above 2 km height. The trajectory for the arrival height of 2.5 km crossed fire areas at heights well within the continental boundary layer. Fire smoke uptake was possible during almost two days. The contribution of African smoke and haze to particle extinction at 532 nm reached values around 50% in the upper part of the SAL (Ansmann et al., 2017). In the respective lidar data analysis we make use of the extinction retrieval by means of the Raman-lidar method (Groß et al., 2015; Haarig et al., 2017a) and compare the obtained total particles extinction coefficient and with the sum of the contribution of dust and non-dust extinction coefficients to the total particle extinction coefficient. And this comparison reveals that the non-dust component must be an aerosol with a high extinction-to-backscatter ratio of around 50 sr which is typical for continental fine-mode urban haze or fire smoke. More details to the extinction comparison and use of the Raman lidar method in this context can be found in Mamouri and Ansmann (2017).

Compared to case 1, the daily mean AERONET Ångström exponent shows slightly enhanced values of 0.3 which may be an indication for the presence of an external mixture of dust and fine-mode particles of continental origin, but may also shows the increasing influence of marine aerosols in the MAL on the AOT with increasing distance from Africa.

The dust layer on 9-10 May (case 3) also potentially contained smoke according to the backward trajectory for the SAL center height of 2.5 km. The respective air mass crossed fire places at heights within the boundary layer over Africa. As discussed below, and in detail in part 2 (Ansmann et al., 2017), the smoke-related light extinction contribution was roughly 20% in the SAL in this case.

The aged dust plume observed on 5 May (case 4) monotonically descended from heights above 4500 m over Africa to 1-2 km height at about 55 °W. The profile of the particle depolarization ratios in Fig. 4 (case 4) shows again lower values than the pure dust value of 0.3, and thus indicates the potential presence of continental smoke and haze. Our analysis suggests again a 20%

contribution to light extinction by continental aerosol pollution. The air masses crossed fire places 9–10 days before arrival over R/V Meteor.

As can be seen in Fig. 4, the particle extinction coefficients for 355 and 532 nm ranged from about 50-100 $Mm^{-1}$ in the SAL over the remote Atlantic Ocean for the moderate dust outbreaks (cases 2–4). Values up to around 300 $Mm^{-1}$ were found

in case 1. A systematic decrease of the SAL backscatter and extinction values with distance from Africa (1700–4300 km) is not obvious from cases 2–4. The found decrease of the SAL AOT is related to the decrease of the SAL vertical extent. The 500 nm AOT decreased from 0.7 (case 1), over 0.3 (case 2) and 0.18 (case 3) towards about 0.15 (case 4). If we subtract a mean marine AOT of around 0.05 and a smoke-haze contribution of 10% (case 1), 40–50% (case 2) and 20% (cases 3-4) to the SAL AOT, the pure dust AOT in the SAL was close to 0.6 (case 1), 0.15 (case 2), 0.1 (case 3), and 0.08 (case 4).

Dust-related particle extinction coefficients in the SAL of 40–80 $Mm^{-1}$ in cases 2–4 and of up to 270 $Mm^{-1}$ in case 1 point to dust mass concentrations of 65–130 $\mu g\, m^{-3}$ and 450 $\mu g\, m^{-3}$ when applying recently updated dust mass–to-extinction conversion factors (Mamouri and Ansmann, 2014, 2017). For freshly emitted dust (over Africa) with a rather low fine dust fraction the conversion factor and thus the estimated dust concentrations may be even 25–30% larger (Osborne et al., 2008). Such young dust plumes may have been observed on 23 May 2013 (case 1). Dust mass concentration profiles are discussed

and compared with respective model forecasts in part 2 (Ansmann et al., 2017).

The 532 nm particle depolarization ratios of >0.25 (case 1), of mostly 0.2–0.23 (case 2, lower part of the SAL), and of about 0.2 (cases 3–4) in Fig. 4 suggest a gradual lowering of the dust contribution to total particle backscattering in the SAL with distance from Africa from >80% in case 1 to values of 60-70% in cases 3 and 4. These dust backscattering fractions of 60% to more than 80% together with the observed particle (dust + non dust) lidar ratios at 532 nm in the SAL of 50–60 sr (case 1,

upper part of the SAL), around 60 sr (case 2, upper part of the SAL), and 40–50 sr (case 3 and case 4, at SAL center height, see Fig. 4) suggest lidar ratios for non-dust particles of around 60 sr (cases 1 and 2) and 30 sr (cases 3 and 4). In this estimation, we assume that the dust lidar ratio is 50–60 sr for western Saharan dust (Groß et al., 2011; Tesche et al., 2011a).

A critical point in our lidar data analysis is the smoke contribution to backscattering and extinction. Müller et al. (2007a) showed that fire smoke particles grow during long-range transport by a number of reasons such as, e.g., gas-to-particle conver-

sion of organic and inorganic vapors during transport, condensation of large organic molecules from the gas phase on existing particles, particle coagulation, photochemical and cloud-processing mechanisms, and hygroscopic growth (Müller et al., 2005; Nikonovas et al., 2015). The surface-area mean radius (denoted as effective radius) of the size distribution of fire smoke was found to increase by a factor of 3 within a travel time of a week, from 0.1–0.15 $\mu m$ to 0.3–0.5 $\mu m$ (Müller et al., 2007a). As a consequence, the extinction-to-backscatter ratio may decrease from, e.g., >60 sr to values <40 sr, as the shipborne lidar

observations are compatible with an aged smoke layer. In fact, lidar ratios around 30 sr were frequently observed at Leipzig, Germany, in outflow aerosol plumes from North America (Müller et al., 2007b), and at the Maldives in aerosol layers advected from rural areas with high biomass burning activity of central-southern India (Franke et al., 2003).

In the discussion of the non-dust contributions to the SAL backscatter and extinction coefficients, the influence of marine particles causing 532 nm lidar ratios of 20–25 sr (Groß et al., 2011; Dawson et al., 2015; Haarig et al., 2017b) cannot fully be

ruled out. During periods with stronger trade wind cumulus convection, cloud tops may partly penetrate into SAL base and

inject marine aerosol particles during the updraft phases. Another source for marine particles is related to sea breeze events at the African west coast associated with the potential injection of marine aerosol into the dust layer when the dust outbreak plumes move westward and cross the coastal areas of West Africa. However, the data presented here are insufficient to either support or deny such an influence. Airborne in situ aerosol observations (long distance flights) would be desirable to measure the aerosol mixing state, degree of internal mixing, aging effects, and size distribution changes between Africa and America during both, summer (dust) and winter (dust and smoke) transport conditions.

The simultaneous observations of depolarization ratios at 355 and 532 nm allow further interpretation of the SAL aerosol mixing state. The SAL depolarization ratios at 355 and 532 nm decreased with distance from Africa from maximum values close to 0.23 at 355 nm and 0.27 at 532 nm (case 1) to values around 0.2 at both wavelengths (cases 3 and 4). For comparison, maximum dust linear depolarization ratios with values close to 0.25 (355 nm) and around 0.3 (532 nm) were found during SAMUM-1 (Freudenthaler et al., 2009) and SAMUM-2 (Groß et al., 2011). The difference of about 0.05–0.075 between the 532 and 355 nm particle depolarization ratios in cases 1 and 2 decreases to ≈0.02 and almost zero with distance from Africa in cases 3 and 4, respectively. Strongly growing smoke particles can explain the decreasing wavelength dependence. As shown by Müller et al. (2007a) and Ansmann et al. (2009), the increase of mean smoke particle size during long-range travel decreases the Ånsgtröm exponent for aged smoke towards values characteristic for mineral dust. This means that the relative impact of aged smoke on particle backscatter and extinction increases more strongly at 532 nm than at 355 nm with travel time, and as a consequence the depolarization ratio decreases more strongly at 532 nm than at 355 nm.

The backscatter and extinction-related Ångström exponents in the SAL in Fig. 4 are typical for aerosols dominated by desert dust. The Ångström exponent (Ångström, 1964) was originally introduced to describes the wavelength dependence of AOT. In the lidar community, the Ångström exponent is also used to characterize the wavelength dependence of particle backscatter and extinction coefficients. Low values around zero for the short wavelength range from 355–532 nm within the SAL are in agreement with the SAMUM-2 observations (Cabo Verde, summer 2008, Tesche et al., 2011a), and are even consistent with the assumption of a mixture of large smoke particles and Saharan dust (cases 3 and 4). The stronger backscatter wavelength dependence for the 532–1064 nm wavelength range (bsc532/1064), expressed here by an Ångström exponent around 0.8 is also typical for desert dust plumes after leaving the African continent (Tesche et al., 2011a; Haarig et al., 2017a) and reflects the changes in the dust size distribution with a strong decrease of the dust particle number concentration for particles with diameters >5 $\mu$m. Examples of size distributions observed with aircraft over Cabo Verde and Barbados during SALTRACE in the June 2013 are given in Weinzierl et al. (2017).

Although the backscatter-related Ångström exponent is usually $\geq 0$ for the 355–532 nm wavelength range, in rare cases the Ångström exponent is $< 0$ as observed in the center of the dust layer on 23 May 2013 (see Fig. 4, case 1, 2–3 km height range). This finding was already discussed by Kanitz et al. (2014). Veselovskii et al. (2016) recently performed lidar measurements of Saharan dust in Senegal and presented several cases with 532 nm backscatter coefficients significantly higher than the ones at 355 nm. This spectral behavior may be caused by a specific chemical composition of the dust particles (and thus specific refractive index characteristics).

The radiosonde profiles of temperature and relative humidity (RH) in Fig. 4 (cases 2-4) are in consistency with the layering structures as observed with lidar and shown in Figs. 2 and 4. The dust layer is drier and warmer (indicated by a strong temperature inversion at SAL base) than the surface-near layers (MBL, MAL). The temperature increased by 6–7 K within 150 m at SAL base in case 2. The less sharp boundary between MAL and SAL (in terms of temperature and RH) in cases 3 and 4 is probably the result of an increasing impact of cloud processes and vertical mixing with increasing travel time over the ocean. The steady increase of the RH with height in the SAL (in cases 1, 3, and 4) indicates well-mixed conditions in the SAL. The RH profile in cases 3 and 4 show a two-layer structure. The lower layer is the cloud-free part of the MBL, the sub-cloud layer (Siebert et al., 2013), and the upper layer is the cloud layer, i.e, in our notation the upper part of the MAL from cloud base of the forming trade wind cumuli up to the trade wind inversion (SAL base).

Figure 6 finally provides an overview of the layer mean optical properties for 532 nm, separately for the MAL and SAL. In 20 nights within the period from 1–23 May, Raman lidar observation over extended time periods of clear skies were possible. Only on 15–16 and 16–17 May continuous occurrence of low level clouds prohibited Raman lidar observations (see Fig. 1, center panel). Until 19 May 2013, the MAL top height was >1 km so that a proper separation of the MAL and SAL optical properties from the lidar profile data was possible. The minimum measurements height was generally set to 400 m. In this way, we avoided a potential bias in the results caused by uncertainties in the correction of the incomplete laser-beam RFOV overlap. Vertical signal smoothing effects close to the layer boundaries are considered in the calculations of the layer mean values by using only data sufficiently above layer base and below layer top. During the last days of the cruise, however, the MAL top height ranged from 250–750 m as shown in Fig. 1 and prohibited an accurate determination of the MAL optical properties (in the near-range of the lidar). For this final period of the cruise (20–23 May period), lidar data integration in the lowest part of the atmosphere always covered the height range from 400 to 900 m to keep the signal-noise-related uncertainties in the data analysis sufficiently low. The respective MAL values shown in Figure 6 are thus influenced by dust occurrence.

The findings in Fig. 6 can be summarized as follows: The MAL mean 532 nm backscatter and extinction coefficients vary strongly, from 1–7 $Mm^{-1}$ $km^{-1}$ and 25-150 $Mm^{-1}$, respectively. This is related to the changing weather and wind-stress conditions which control the amount of sea salt particles in the air. Because of the strong backscatter efficiency of marine particles, the marine backscatter coefficients are typically much larger than the SAL dust backscatter values. Such a strong difference between MAL and SAL data is not visible in the case of the extinction coefficients. The SAL mean extinction values were mostly found between 40 and 100 $Mm^{-1}$ (on average 68±37 $Mm^{-1}$ for all 20 nights). However, during strong dust outbreaks as the 23 May case in Fig. 6b the values can be much higher. A steady decrease of the SAL mean extinction coefficient with travel time is not visible. The SAL (dust + non-dust) AOT in Fig. 6 (bottom panel) ranged from 0.02 to 0.2 over the ocean, more than 1000 km west of the African coast. The 20 night average of the SAL AOT is 0.15±0.13 at 532 nm.

Our shipborne observations of the SAL mean extinction coefficients are in good agreement with the SAMUM-2 observations at Cabo Verde (four weeks in May-June 2008) and the SALTRACE observations (Groß et al., 2015; Haarig et al., 2017a). The 532 nm dust extinction coefficients were mostly in the range from 30–150 $Mm^{-1}$ (on average about 80±60 $Mm^{-1}$) and 50-150 $Mm^{-1}$ in the SAL over Barbados in the summers of 2013 and 2014. Carlson (2016) presented SAL mean dust extinction coefficients of the order of 200 $Mm^{-1}$ at 500 nm wavelength and thus about a factor of two higher values. The extinction values

of Carlson (2016) are obtained from a correlation between the total (MAL + SAL) AOT and the SAL layer depth, estimated from radiosonde profiles of meteorological parameters. Thus the SAL AOT is overestimated by the marine AOT contribution and the SAL geometrical depth is obviously underestimated when taking meteorological data and detailed aerosol backscatter profiles obtained with lidar are not available. In the study of Carlson (2016) the mean SAL base over Barbados was at 750 hPa (about 2500 m) and the mean SAL top at 614hpa (4500m). Groß et al. (2015) found the SAL base and top heights usually between 1500 and 2000 m and 4000-4500 m, respectively, during the SALTRACE summer campaign in June and July 2013.

As in the case of the SAL extinction coefficients, the layer mean lidar ratios (on average $43\pm8$ sr) and depolarization ratios (on average $0.19\pm0.09$) in Fig. 6 also indicate a travel-time-independent dust optical properties and suggest homogeneous aerosol conditions (regarding particle size spectrum and aerosol mixture) over the Atlantic. A SAL mean particle linear depolarization ratio of 0.17 to 0.25 at 532 nm suggests smoke or sea salt contributions to the backscatter coefficient of 20% to 50%. In terms of light extinction, the relative smoke contribution is smaller. For a lidar ratio of 30 sr for aged smoke and a lidar ratio of 50 sr for dust, the smoke impact reduces to 10-30% for the extinction coefficients.

The ship cruise allowed us also to describe clean marine conditions in terms of lidar specific optical properties at sites far away from continents. The MAL lidar ratios (10-25 sr at 532 nm, mean of $17\pm5$ sr) and depolarization ratios (0.01–0.04, mean of $0.025\pm0.015$) show typical values for clean marine conditions until 19 May (Groß et al., 2011), i.e., when excluding the observations from 20–23 May. The mean MAL extinction coefficient was $67\pm45$ Mm$^{-1}$ for the 1–19 May period.

### 3.3.1 Conceptual model versus lidar observations (part 2)

As mentioned at the end of Sect. 3.2.1 and emphasized in Fig. 3, according to the conceptual model gravitational settling is widely responsible for the removal of dust from the SAL in the absence of clouds and precipitation. Wind-shear-inducted turbulent downward mixing in the SAL base region may additionally contribute to dust removal. However, several observational studies indicate that the role of sedimentation is overestimated in the conceptual model as well in state-of-the-art dust transport models as is discuss in detail in part 2 (Ansmann et al., 2017).

Kim et. al. (2014) found a decrease of the dust load in terms of AOT by a factor of two from the African coast downwind to Barbados according to satellite observations, whereas five dust transport models produced a decay by a factor of roughly 4–10 from the west coast of Africa to 60°W. Furthermore, a retention of the coarse mode particles was observed in recent Saharan dust studies by Ryder et al. (2013), van der Does et al. (2016), and Weinzierl et al. (2017). By comparing their airborne SALTRACE observations over Cabo Verde and Barbados, Weinzierl et al. (2017) showed that the particle number concentration for coarse-mode particles with diameters $>10$ $\mu$m was reduced by only 60%. According to their gravitational settling computation SAL dust particles with diameters $>7$ $\mu$m should not be able to reach Barbados.

The question is now: what processes can weaken the gravitational settling effect? Gasteiger et al. (2017) argue that absorption of solar radiation introduce turbulent mixing of dust within the SAL and thus upward and downward transport of dust which weakens the pure sedimentation-based dust removal effect. Colarco et. al. (2003) and Yang et al. (2013) discuss the impact of different shapes of dust particles on falling speed and gravitational settling behavior. Ulanowski et al. (2007) observed that dust layers have an impact on the atmospheric electric field, and argue that dust particles can become charged (when colliding

with themselves or the underlying surface),and may be vertically aligned in the electric field, and conclude that these charging effect influence the downward transport of dust. Radiosonde wind profiles always show variations in wind direction and wind velocity throughout the SAL vertical column and thus indicate the potential for some wind-shear-induced turbulent mixing (including upward motion of dust particles). Airborne Doppler lidar observations of vertical and horizontal wind components

in the SAL over the open Ocean during SALTRACE 2013 corroborate this hypothesis (Chouza et. al., 2016a, b).

Our observations are in accordance with these findings. If particle sedimentation would be dominating in the SAL, we should observe a decrease of the coarse dust fraction with height, i.e., an accumulation of the larger dust particles in the lower part of the SAL after dust transport over days and distances of 4000 km and more (Gasteiger et al., 2017). And this decrease of coarse dust fraction with height should then be reflected in the height profile of the particle depolarization ratio and other optical

parameters such as the particle extinction coefficient and the backscatter Ångström exponent. As explained in Sect. 2, coarse dust should lead to depolarization ratios of 0.35-0.4 at 532 nm, whereas fine dust should exhibit causes depolarization ratios below 0.2. Thus, according to the conceptual model we should observe a systematic decrease of the depolarization ratio with increasing height in cases 3 and 4 from SAL base to top. But this is not found. No systematic and significant decrease of the depolarization ratio and of the extinction coefficient and an increase of the backscatter Ångström exponents were observed.

Our findings are in agreement with the lidar dust studies over the Atlantic by Yang et al. (2013) and Haarig et al. (2017a).

Furthermore, the measured particle extinction coefficients range from 50–100 $Mm^{-1}$ in the SAL and the dust extinction values between 40 and 80 $Mm^{-1}$ for cases 2–4. Again, a systematic decrease of SAL extinction values with increasing travel time is not observed. Also, no trends in the layer mean values of the lidar ratio and the depolarization ratio in Fig. 6 are visible **which supports the hypothesis that size-dependent gravitational settling has only a weak impact on the dust amount in**

**the SAL and significant changes in the dust size distribution with increasing transport time do not occur.**

## 4   Conclusions

During a one-month transatlantic cruise from Guadeloupe to Cabo Verde over 4500 km the aerosol layering structures over the tropical Atlantic were continuously monitored with a multiwavelength lidar aboard the German research vessel R/V Meteor. The lidar allowed us to retrieve a rich set of SAL optical properties during an early summer period in the final phase of the

biomass burning season. Dust removal aspects could be studied in large detail over the remote Atlantic Ocean. We observed a clear aerosol vertical layering (MBL, MAL, SAL) during the west-to-east cruise. Good agreement with the model was found regarding the changes in the SAL base height (increase) and top height (decrease) with distance from Africa as described by the conceptual model (Karyampudi et al., 1999). The shipborne observations suggest a mixing of smoke and dust within the SAL during the late phase of the biomass burning season in central-western Africa. Dust-smoke mixing aspects are not covered

by the conceptual model. We further concluded that the removal of dust from the atmosphere below the SAL must be very efficient, in agreement with the conceptual model. On the other hand, our observations corroborate previous studies that the removal of dust from the SAL is less efficient than expected by the model, obviously as a consequence of less efficient particle

sedimentation. Besides gravitational settling other processes must be active to prolong the lifetime of dust in the SAL. The discussion will be continued in part 2 (Ansmann et al., 2017).

The SAL vertical mean depolarization of 0.19±0.09 and lidar ratio of 43±8 sr for the May 2013 observational period were clearly lower than the respective values for pure dust of around 0.3 and 55 sr. 80–90% of the 532 nm particle extinction coefficient in the SAL was estimated to be caused by dust particles in most cases. The further analysis of the SAL lidar ratios suggest that the smoke may have grown during the long-range transport and changed its backscattering and extinction properties so that the lidar ratio decreased considerably from values around 60 sr (fresh smoke) to values close to 30 sr after a travel time of 5 to 10 days over the tropical Atlantic. Since the conceptual model only covers the summer mode of dust transport, an extension towards including winter mode conditions with complex mixing of dust and smoke at lower altitudes would be desirable.

The shipborne lidar observations in May 2013 fit well into the dust characteristics and layering structures gained from the SAMUM and SALTRACE field campaigns regarding the long range transport of dust. Good agreement regarding the dust optical properties was found when comparing the dust measurements in Morocco, on Cabo Verde, and Barbados, and aboard the R/V Meteor across the Atlantic.

The ship cruise also provided ideal conditions for the measurement of pure marine aerosol optical properties far away from disturbing continents. The results are consistent with the SAMUM-2 observations on Cabo Verde and the winter SALTRACE observations at Barbados. During both campaigns a few days with pure marine conditions occurred. During the ship cruise marine, dust-free conditions in the lowest 1000–1500 m of the atmosphere prevailed continuously for more than two weeks. Typical marine lidar ratios were found to be 15-20 sr at 532 nm. The marine depolarization ratios (controlled by wet sea salt particles) accumulated around 0.03.

In the companion paper Ansmann et al. (2017), the shipborne lidar observations, discussed in this part 1 predominately in terms of dust optical properties, are further analyzed to quantify the fine dust, coarse dust, and non-dust contributions to light extinction and mass concentration by means of the POLIPHON method (Mamouri and Ansmann, 2014, 2017). The lidar products are compared with respective forecasts of a regional and two global dust models. Highlight of part 2 is the distinct comparison of observed and modeled fine and coarse dust profiles and thus to focus on aspects regarding the modeled dust size distribution and changes during long-range transport.

The observations in this part 1 and in the follow-up article (part 2) as well as all the advanced lidar observations performed during the dust-related field campaigns in the last 10–15 years clearly indicate the importance and need for comprehensive vertically resolved dust measurements to better understand the life cycle of atmospheric dust, and to improve atmospheric dust modeling from emission to deposition, the interaction of dust with the radiation field, and the dust impact on cloud formation and precipitation. The built-up of a permanent ground-based dust-monitoring lidar networking infrastructure would be a big step forward to support dust life cycle and impact research and dust forecasting by combining the continuously available lidar observations with dust modeling efforts.

To support and confirm our lidar-based findings (obtained by using many assumptions), east-west long-distance research flights within the SAL across the tropical Atlantic would be very helpful with focus of the airborne in situ aerosol observations

on the aerosol mixing state, fractions of internally and externally mixed aerosols, dust aging caused by cloud processing and chemical aging, changes in the dust size distribution between the Saharan dust source and the Caribbean, South and North America, and also aging of biomass-burning smoke particles in the dust-smoke mixtures during the biomass burning season. Research flights are required during both summer (dust) and winter (dust and smoke) half years. Further points of airborne research should cover aerosol-cloud interaction aspects, i.e., CCN and INP characterizations across the Atlantic with the specific question regarding the contribution of dust and non-dust particles to the CCN and INP reservoirs and how dust aging changes the ability of dust to serve as CCN and/or INP. First steps into this complex dust research field were performed during the SALTRACE campaign (Weinzierl et al., 2017).

## 5 Data availability

Radiosondes for measuring, temperature, humidity, and wind profiles were regularly launched on board the research vessel at noon and midnight UTC by the German Weather Service (AWI, 2016). GDAS (Global Data Assimilation System) height profiles of temperature and pressure of the National Weather Service's National Centers for Environmental Prediction (NCEP) are used, in addition, for our computations of Rayleigh scattering contributions (NOAA's Air Resources Laboratory ARL, https://www.ready.noaa.gov/gdas1.php) (GDAS, 2016). The shown AOT data are available at *http://aeronet.gsfc.nasa.gov/new_web/cruises_r* (MAN, 2016). The Maritime Aerosol Network (MAN) is a component of AERONET (Smirnov et al., 2009). The trajectories are calculated with the NOAA (National Oceanic and Atmospheric Administration) HYSPLIT (HYbrid Single-Particle Lagrangian Integrated Trajectory) model (http://ready.arl.noaa.gov/HYSPLIT_traj.php) (HYSPLIT, 2016) using GDAS meteorological data (Stein et al., 2015). In addition, fires detected by MODIS (Moderate Resolution Imaging Spectroradiometer) on board the Terra and Aqua satellites are used and are avaialble at http://rapidfire.sci.gsfc.nasa.gov/firemaps (MODIS, 2016). The lidar are available at TROPOS. Please contact Ronny Engelmann (ronny@tropos.de) for further questions.

*Acknowledgements.* We thank the R/V Meteor team and German Weather Service (DWD) for their support during the cruise M96. We appreciate the effort of AERONET MAN to equip research vessels with sun photometers for atmospheric research. We acknowledge the time and effort that the anonymous referees put in the review. Their numerous suggestions improved the paper significantly.

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

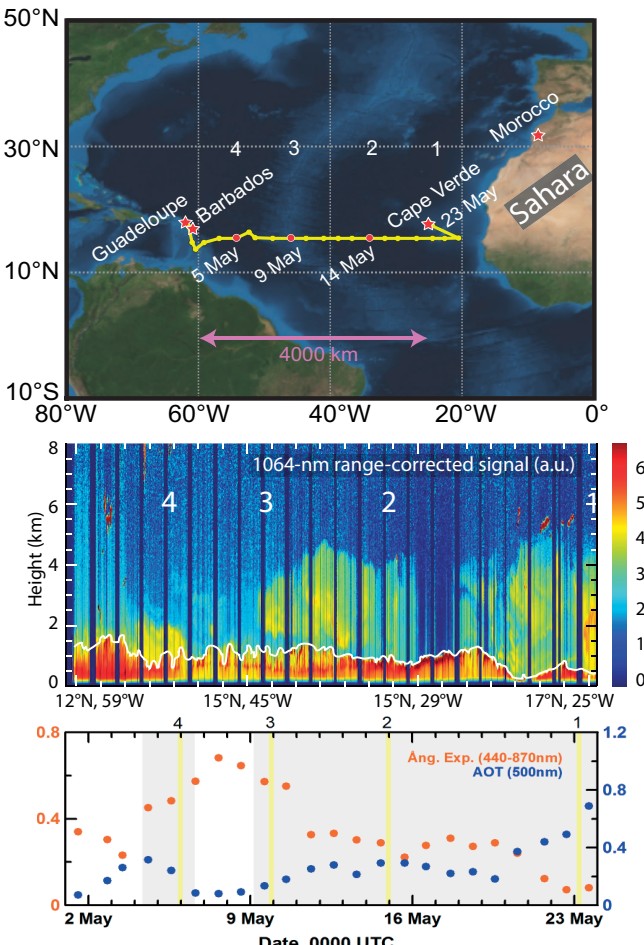

**Figure 1.** (Top) Cruise track of the R/V Meteor from Guadeloupe (29 April 2013) to Cape Verde (23 May 2013) plotted as a thick yellow line (Kanitz et al., 2014). The SAMUM-1 (Morocco), SAMUM-2 (Cape Verde), and SALTRACE (Barbados) field sites are marked by red stars. Red circles and dates (5, 9, 14 and 23 May) indicate the locations of four lidar observations (cases 1, 2, 3, and 4) discussed in detail in this paper and the follow-up article. (Center) Dust-free marine aerosol layer (MAL, top height as a white line) and lofted Saharan air layer (SAL). The composite is based on lidar measurements of the range-corrected 1064 nm backscatter signal. Measurement breaks around 1200 local time (dark vertical lines) are due to high sun elevation and shut down of the lidar. (Bottom) Time series of daily mean sun photometer observations of aerosol optical thickness (AOT) (blue circles) and Ångström exponent (orange circles). The gray-shaded areas indicate time periods with lofted dust. The yellow vertical lines indicate the selected four observational cases.

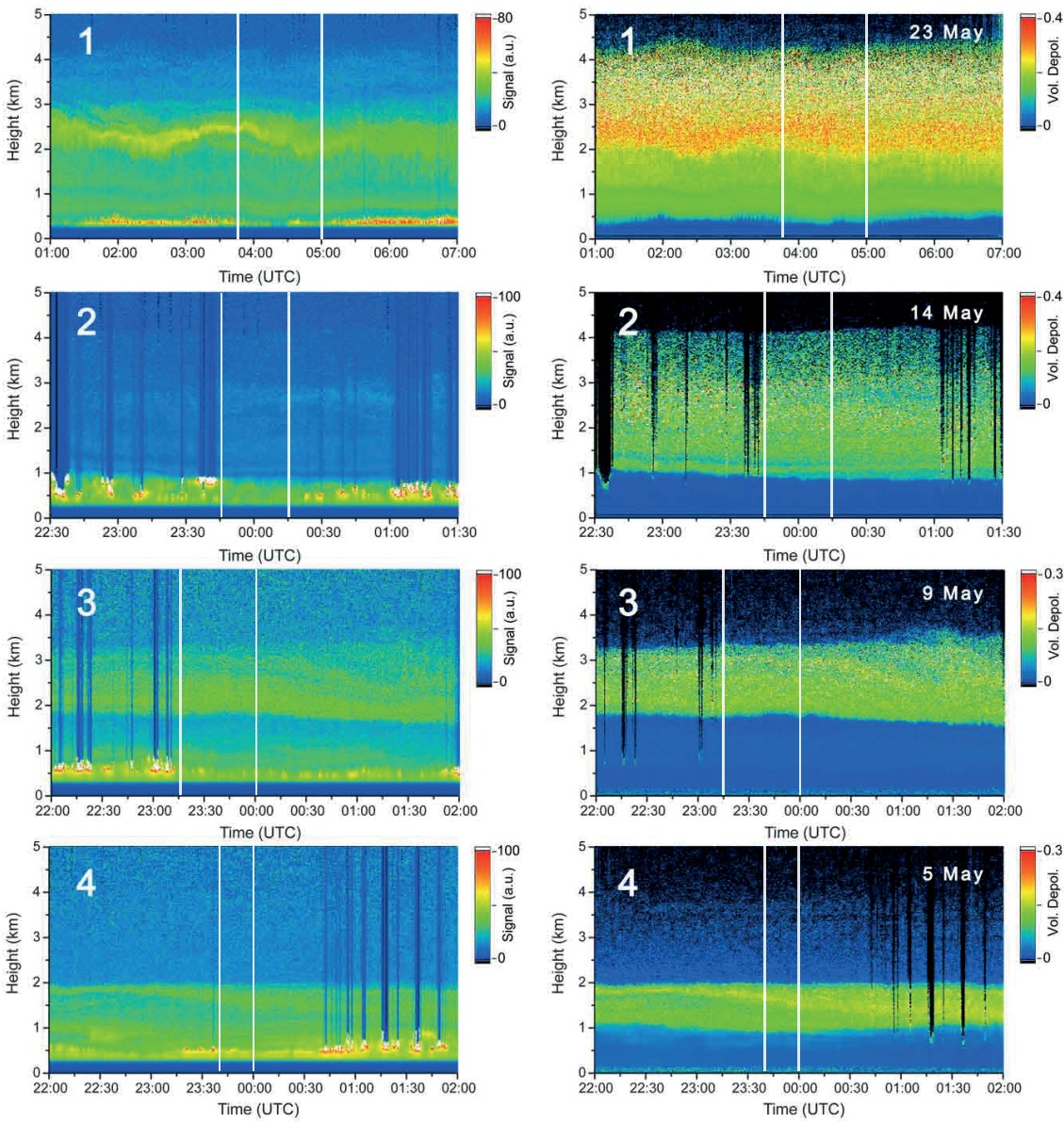

**Figure 2.** Marine and dust layers over the tropical Atlantic about 1000 km (1), 1700 km (2), 3300 km (3), 4300 km (4) west of the African coast. The 532 nm range-corrected signal (left) and 532 nm volume linear depolarization ratio (right) are shown. Linear color scale is used. Temporal and vertical resolution is 30 s and 7.5 m, respectively. The SAL is on top of the MAL (low depolarization ratios displayed in blue in the right panels). White spots in the left panels (below 1 km) indicate trade wind cumuli close to the top of the MBL, the convective part of the MAL. MBL top is at 400–500 m (case 1), at about 1 km (case 2), 600–900 m (case 3), and 600–1000 m (case 4). Vertical lines indicate the signal averaging periods for which profiles of optical properties are discussed in Sect. 3.3. Lidar overlap effects (lowest 250 m) prohibit a clear aerosol detection in the near-range of the lidar in the left panels.

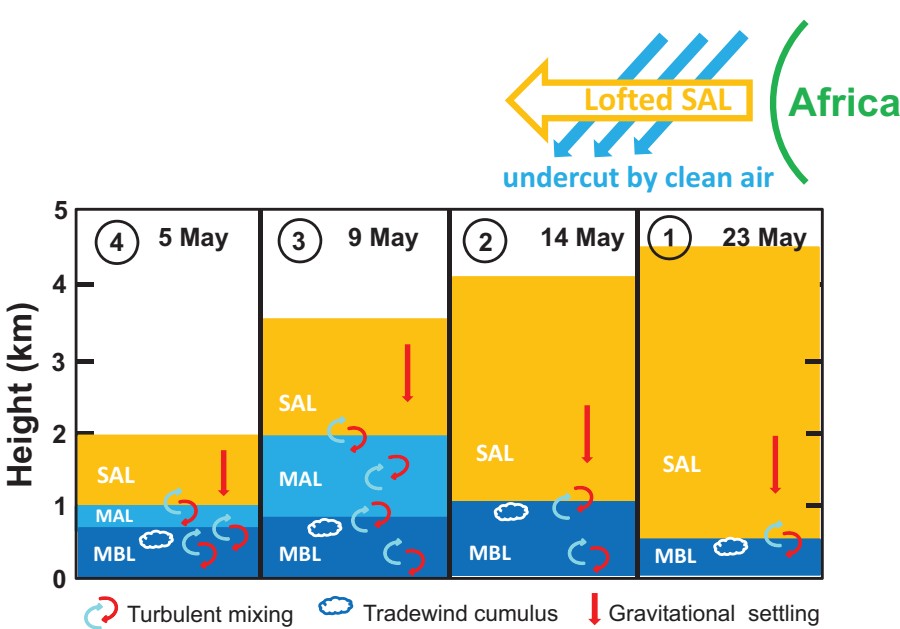

**Figure 3.** (Top) Sketch of the east-to-west dust transport in the lofted SAL which is undercut by clean trade winds from the Northeast (occurring in the MAL) according to the dust plume conceptual model of Karyampudi et al. (1999), described in Sect. 1.1. (Bottom) Sketch of dust layering (SAL in orange, MAL in blue, the convective part of the MAL, i.e., the MBL, in dark blue). Base and top heights of the MBL, MAL, and SAL are taken from Fig. 2. Gravitational settling (symbolized by a red arrow) is responsible for the removal of dust from the SAL, turbulent downward mixing (symbolized by curved arrows) plays an important role in the removal of dust from the MAL according to the model.

.

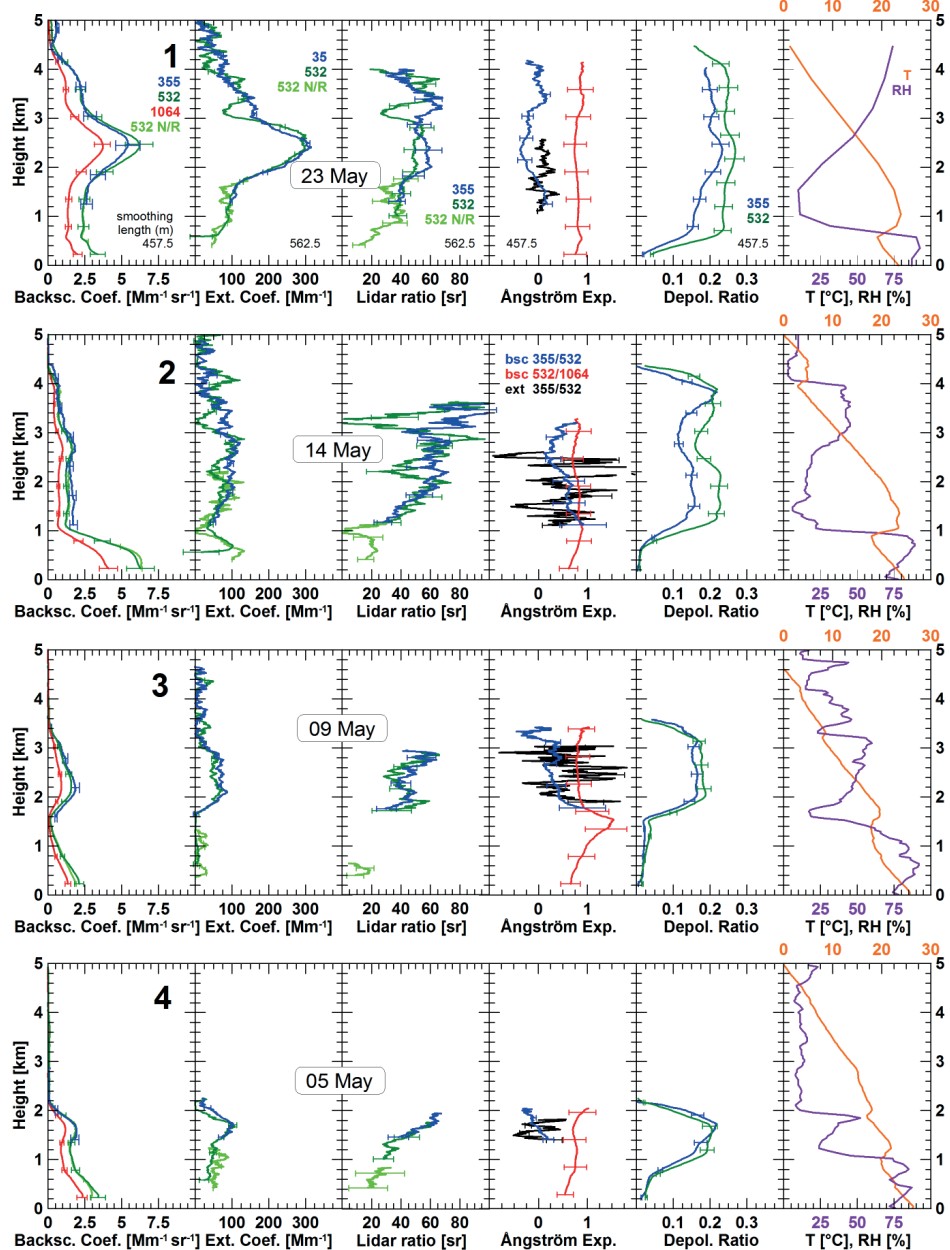

**Figure 4.** Profiles of the particle backscatter coefficient at 355, 532, and 1064 nm, extinction coefficient, extinction-to-backscatter ratio (lidar ratio), and particle linear depolarization ratio at 355 and 532 nm, and backscatter-related (bsc) and extinction-related (ext) Ångström exponents. Temperature and relative humidity (from radiosonde, except 23 May, GDAS) are given in addition. Mean profiles of the optical properties for the time periods on 23 May 2013, 03:45–05:00 UTC (case 1), 14–15 May 2013, 23:45–00:15 UTC (case 2), 9 May 2013, 23:15–24:00 UTC (case 3), and 5 May 2013, 23:40–24:00 UTC (case 4) are shown. The label 532 N/R denotes the 532 nm near-range receiver channel. The vertical signal smoothing length for the profiles of backscatter coefficient and particle linear depolarization ratio is 457.5 m, the rest is smoothed with 562.5 m window length. The signal averaging periods are indicated by white vertical lines in Fig. 2.

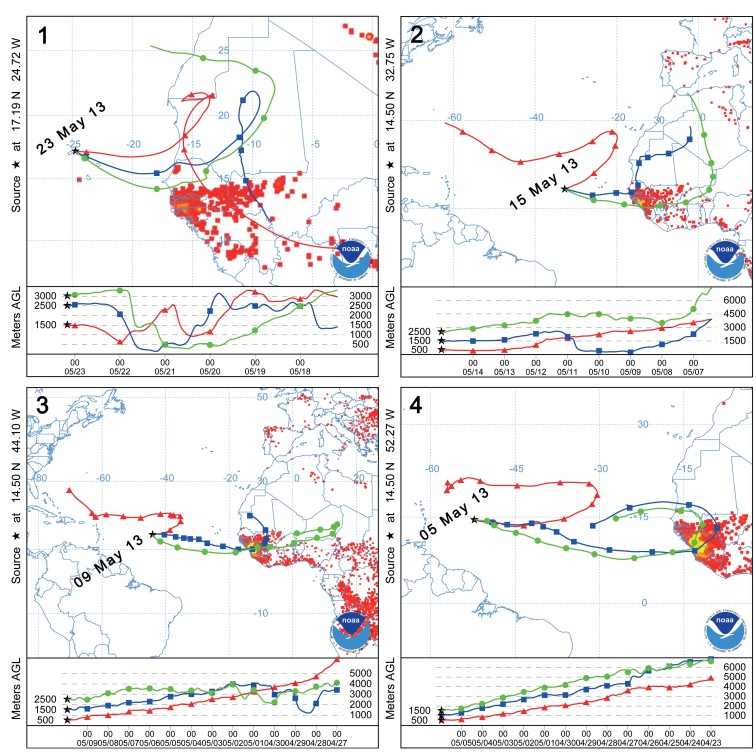

**Figure 5.** Five-day to 13-day HYSPLIT backward trajectories for 23 May 2013, 04:00 UTC (case 1), 15 May 2013, 00:00 UTC (case 2), 9 May 2013, 23:00 UTC (case 3), and 5 May 2013, 23:00 UTC (case 4). Symbols indicate air mass transport from day to day. The arrival height level of 500 m (red) is in the MAL. Arrival heights of 1500–3000 m (blue, green) are in the SAL. In addition, fires (red dots) detected by MODIS on board the Terra and Aqua satellites are shown accumulated over a 10-day period each (21–30 April 2013 for cases 3 and 4, 11–20 May 2013 for case 2, and 21–30 May 2013 for case 1).

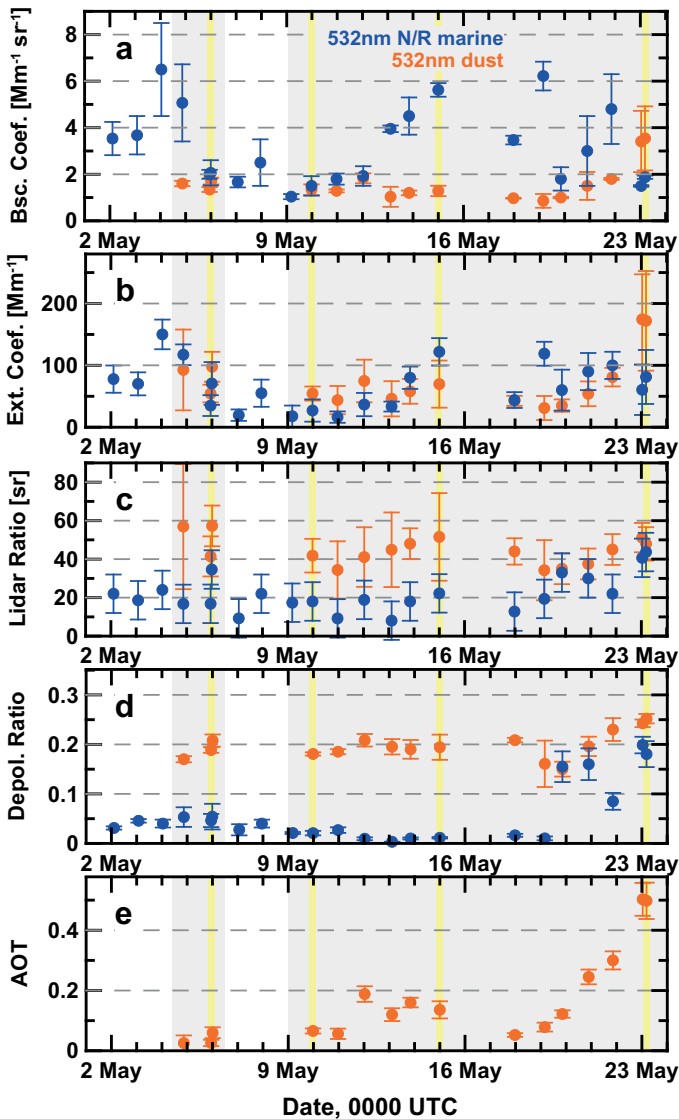

**Figure 6.** Layer mean values of (a) 532 nm particle backscatter coefficient, (b) extinction coefficient, (c) lidar ratio, and (d) linear depolarization ratio observed with lidar in 20 nights in May 2013. Values for the SAL are given in red. The blue circles show the values for pure marine conditions within the MAL (1-19 May) and for mixtures of dust and marine particles in the lowest 900 m (19 May evening to 23 May morning). 30–120 minute averages are presented. Cases 1–4 are indicated as vertical yellow lines. The SAL AOT at 532 nm is given in panel e. Error bars (one standard deviation) indicate the retrieval uncertainty and atmospheric variability within the analyzed layers. The two periods with dust are shown as gray areas.