# Peer review of "Profiling of Saharan dust from the Caribbean to West Africa, Part 1: Layering structures and optical properties from shipborne polarization/Raman lidar observations"

_Atmospheric Chemistry and Physics, 2017_

## Referee Comment (RC1) · Anonymous Referee #1 · 28 Mar 2017

In April-May 2013 the R/V Meteor did a ∼4000 km long transatlantic cruise from Guadeloupe to Cape Verde, carrying a PollyXT lidar which took continuous measurements of the dust vertical structure and optical properties. A hand-held sunphotometer was also used (Microtops), and daily radiosondes were launched. The main datasets and results are published in a GRL paper, together with some elements of meteorological interpretation with COSMO-MUSCAT and HYSPLIT (Kanitz et al, 2014). That paper highlighted the properties of fresh and aged dust, and characterised it in terms of Angstrom exponent, lidar ratio, particle depolarization ratio, aerosol optical depth, and vertical distribution.

[Figure]

The paper by Rittmeister et al builds on that dataset to conduct further analysis of the results. In particular, the optical properties of the marine aerosol are briefly investigated in addition to dust, and four vertical profiles distributed along the ship's track are investigated in more detail in terms of structure, temporal evolution, and airmass trajectories. A conceptual model is discussed, and a nice sketch of the atmospheric layering across the Atlantic is given (Figure 9). A further follow-on paper is announced within the article text, where more detailed derivations will be shown, including the mass concentrations of fine and coarse dust, and their comparison with aerosol transport simulations.

I have a few concerns with the paper. There is a very large degree of overlap with the previous paper by Kanitz et al, with several figures in common as well as repeated information given in the text, and I think that the paper would really benefit from highlighting new findings rather than going again through material already covered in that paper. Moreover, I believe that the presentation of the material would benefit from a major rewrite. I find that the structure of the paper is not always optimal and that some statements are given as granted whereas a justification or a reasoning could be highlighted. Finally, as a follow-on paper is foreseen, I would recommend submitting those two at the same time and with an organic plan. Possibly, after trimming large parts of the present manuscript as I suggest, they could be combined in a single paper.

This is promising research, and it is not light heartedly that I recommend the rejection of the paper at this stage. I believe that it will benefit from a careful revision and restructuring, and I will be happy to review it again, if it is resubmitted along the lines that I suggest.

MAJOR POINTS:

1) Title: This title does not highlight what is new in this paper compared to the previous paper by Kanitz et al. Moreover, if the plan is to have two papers I would encourage submitting them as Part 1 and Part 2 more or less at the same time.

2) Abstract: As for the title, the abstract in the current form does not highlight what is

new in this paper, and describes findings which are similar to those of the Kanitz et al paper. The first four lines drive the reader to believe that this is the first description of this transatlantic lidar transect, whereas this is untrue. Moreover, the findings that are reported in the abstract (lidar ratio and depolarization ratio) add very little. Both title and abstract should focus on new findings.

3) Several figures are simply reproductions of figures in Kanitz et al (2014). I suggest to omit them here, and write the paper around the new findings instead. In the specific: Figure 1 corresponds to Figure 1 of Kanitz et al; Figure 2 corresponds to Figure 2b+c of Kanitz; Figure 3 corresponds to Figure 2a of Kanitz. Moreover, Figure 6 first and last rows are very similar (although not identical) to the data in Figure 3 of Kanitz et al.

4) Many findings are presented as new findings and discussed at length, whereas they are instead previous findings from Kanitz et al. Section 2 also does not describe much new material compared to that paper. I would replace the current section 2 with a short summary of the findings by Kanitz et al (not longer than the abstract of that paper). This can then also permit to reduce some of the reproductions of those findings in the current version of section 3.

5) P3 L30: It is true that a shipborne East-West lidar study has never been performed before 2013, but this has been reported before. This is therefore not the highlight of the present paper. I suggest instead to use an approach like the one on P5 L19: "Kanitz et al (2014) already provided an overview and first results".

6) P5 L21-22: This interpretation of the lidar data to indicate a MBL and a MAL comes as a surprise here as no reasoning behind it is given. With some experience of lidar signals I can easily recognise where the SAL and the MBL are (with help of the depolarization plot) but you cannot assume that all readers have this knowledge. The MAL is an new concept to me, I fail to see it at a glance in this plot, and I have not found a convincing explanation in the paper of how the data presented prove its existence.

7) There are other interpretations of the data given with little explanation in the text;

some of them are listed below (e.g. P5 L29, P6 L8-9, P6 L12-13, P6 L25 and L26, P7 L5-10, P7 L19, P7 L30, P8 L7-8, P8 L34, P9 L3-5, etc.).

8) The "conceptual model" presented in section 4 (P9 L12-32) should really be explained in the introduction.

9) Conclusions: Only 10 lines of conclusions for this work? This is the most important part of the paper. Here you could tie your results with previous research (which you already indicated in the introduction), discuss the caveats and implications, suggest further research, etc. Here is where you justify the benefits of this research in a wider scientific context.

OTHER IMPORTANT POINTS TO CONSIDER:

10) P1 L19: dust as "surface-near" plumes because it is part of the BL: please note that the Saharan BL can be extremely deep (up to 5-6 km in a summertime afternoon); therefore this description is inaccurate.

11) P1 L21: I would think that dust lifted to tropopause height is not as common as the paper describes it; most frequently dust plumes are encountered between the surface and 5-6 km.

12) P2 L1: why do you describe dust as "omnipresent"? Although dust is abundant, I would not think that it is found everywhere (indeed most atmospheric layers around the globe are dust-free).

13) P2 L21: the distance from the W coast of Africa to the Caribbean is about 6000 km; therefore 10,000 km sounds a bit large. The same observation applies to P3 L3 where a distance of 5-8,000 km is indicated.

14) P2 L27-30: the change of topic from dust to smoke is a bit sudden at this point in the paper.

15) P4 L15: between "are available" and "the marine boundary layer" you could add

"from 250 m (full overlap) to XXXX m (limited by SNR), covering ..."

16) P4 L24: you mention separating dust from other aerosol based on the particle depolarization ratio (I suppose this is the method by Tesche et al, 2009) and you base this on an assumption that the depolarization ratio is 0.3 for pure dust. I would challenge this, as ageing along the Atlantic path could change the depolarization ratio of the dust component (and you confirm this fact in this paper actually). I would therefore recommend to take ageing into account when applying this method.

17) P5 L8: I suggest to specify that the radiosondes were launched from the same ship.

18) P5 L13: I recommend to use a word like "prediction" or "computation", because "tracking" usually refers to remote sensing observations (radar, satellite, etc.)

19) P5 L15: I recommend to say "used in conjunction" rather than "combined", unless a new aggregate product has been designed that combines both.

20) P5 L29: The colour scale in the plot indicates the magnitude of the range corrected signal, and NOT the different layers. The attribution of the SAL to "green and yellow colours" (i.e. range corrected signal between 3 and 4.5) is an interpretation, and as such I think it deserves an explicit explanation in the text.

21) P6 L8-9: This statement is substantially correct, but it is not formulated in a useful manner; it has to be clear that it is our interpretation that an AOT of 0.05 corresponds to a dust-free pure marine condition, and that 0.7 corresponds to a major dust outbreak. The lidar data, the backtrajectories, and correct wording can help support this statement.

22) P6 L10: AOT is up to 0.7 (not 0.3)

23) P6 L12-13: larger Angstrom exponent is indicative of smaller particles (and viceversa), not of a given aerosol type. The suggestion that this indicates sea salt or dust is an interpretation, and should be presented as such. In particular, it is reasonable

that as dust travels away from source (as presented in this paper), larger particles undergo deposition and therefore the Angstrom exponent increases but the aerosol type remains "dust". This needs probably to be clarified and accounted for.

24) P6 L14: I appreciate the effort in rationalising what is observed, but before calling the four lidar sections "key stages" I believe that some explanation and discussion could be useful.

25) P6 L16-18: It may be worth specifying that this is deduced from backtrajectories. These trajectories pass over hot spots, and therefore are not capable of ruling out a biomass burning component: this could be explicitly discussed.

26) P6 L20: To give dust an age (7-9 days), how do you determine at which point along the trajectory it was lifted?

27) P6 L25: It is unclear how the statement about mass concentrations is justified.

28) P6 L26: How is the MBL top identified from Figure 4 and how is it found different from the dust base height?

29) P6 L31: smoothing window (365 to 458 m): this is in contrast to the figure, where 457.5 and 562.5 are indicated.

30) P7 L3-4: as Rittmeister et al (2017) is not yet published, may I suggest to cite other existing references about the conversion of optical properties to dust mass concentrations? See e.g.

http://onlinelibrary.wiley.com/doi/10.1029/2007JD009551/full
http://onlinelibrary.wiley.com/doi/10.1029/2000JD900319/pdf
http://onlinelibrary.wiley.com/doi/10.1002/qj.777/full

31) P7 L5-10: which analysis showed that smoke does not dominate this airmass? This is not presented at all in this paper. It is definitely not obvious why these fires do not contribute.

32) P7 L12-13: you say that the backscatter wavelength dependence is due to long-range transport. However, in Figure 6 this applies also to case 1.

33) P7 L19: whereas it is reasonable to think that large particles fall out during long range transport, how does the data support this strong statement?

34) P7 L30: besides the potential mixing with marine particles, the decrease of depolarization ratio could also be ascribed to the ageing of dust (removal of larger particles; coating with water and/or other species, etc.)

35) P8 L1-2: I would remove the hard numbers here and limit to saying that larger/smaller depolarization ratios are expected.

36) P8 L4: In my opinion, the intrusion of marine particles in the SAL has not really been demonstrated in this paper.

37) P8 L7-8: To say that the radiosonde data are in agreement with the lidar observations is again to skip a logical step. Whereas it is clear to me what the authors want to say, I would not think that it is correctly formulated, and as such other readers may find this difficult to understand. I think that the correct statement should be that radiosonde profiles show a consistent layering of the atmosphere with the lidar dataset.

38) P8 L10-11: I am not sure I understand this. In Figure 6, we see that the RH is large below the SAL base and is small above the SAL base (if we take the depolarization profile as indication of where the SAL boundary is). In P10 L4-5 you clearly acknowledge the sharp change in depolarization ratio at the SAL base: this should be evidence against these vertical exchange processes.

39) P8 L14-16: Omit.

40) P8 L18: Here you mention 16 analysed cases. These come as a surprise because they were never mentioned earlier in the paper.

41) P8 L26: Cite literature on the marine LR around 20 sr. Many references exist on

lidar ratio of different aerosol types. For example:

http://onlinelibrary.wiley.com/doi/10.1029/2006JD008292/full

http://www.atmos-chem-phys.net/15/3241/2015/

http://www.sciencedirect.com/science/article/pii/S1352231011006108

http://www.atmos-meas-tech.net/6/3281/2013/

42) P8 L32-33: Maybe removing the lower and upper 250 m of the SAL could prevent the fact that smoothing with a $\sim$500 m window introduces information from layers below or above?

43) P8 L34: I think it is really an overstatement to say that a LR of 40 sr "clearly" indicates an impact of marine particles. The LR of dust is very variable depending of source region (see e.g. http://onlinelibrary.wiley.com/doi/10.1002/grl.50898/full ). Moreover, the authors themselves have already acknowledged in this paper that the ageing of dust can reduce its LR.

44) P9 L1-3: Again I believe that the comments on the depolarization ratio are too sharp, and I would moderate them in terms of possibilistic statements.

45) P9 L3-5: I believe once again that the authors have no evidence for saying that in proximity of the African continent there is no MBL. Indeed, models and campaigns indicate that such a layer exists near the coast. A more plausible explanation could be that the large depolarization ratio is due to fall out of large dust particles from the SAL above.

46) P9 L9-10: This concept has been repeated several times throughout the paper, but I am not persuaded by the arguments as already commented. Ageing mechanisms are plausible causes. There is also no need to repeat a same concept so many times.

47) P9 L19: unclear: "except in disturbance".

48) P10 L14: smoke? not discussed much in this paper

49) Figure 1: omit figure as it is part of Kanitz et al. Continents are not clearly visible.

50) Figure 2: omit figure as it is part of Kanitz et al. The caption does not describe the figure, instead it tries to interpret it.

51) Figure 3: omit figure as it is part of Kanitz et al. The gray-shaded areas are hardly visible when this is printed. It is unclear what criteria where used to delimit them. A longitude x-axis would probably be more useful than a time axis.

52) Figures 4-6 and 9-10: I suggest reversing the order of cases 1-4, to reflect the order of the discussion in the text (P6 L16-22). This would also have the benefit to have the Easternmost panel on the right (and the Westernmost on the left) in Figure 9, i.e. like one would see it on a map.

53) Figure 4: Caption does not explain what is shown (RCS and VDR), does not clarify how the MBL and MAL are distinguished. The data within the incomplete overlap should be treated as missing data instead of commenting on the "blue area" at the bottom. An indication of longitude for each case would be useful. Blue areas in the right hand panel indicate low VDR, which is indicative of dust-free layers; they do not directly indicate dust-free layers.

54) Figure 5 caption: why do you say that the 500 m level is always within the MBL and the arrival heights 1500-3000 m are always in the SAL? I suppose this is indicated by the lidar profiles, but if it is the case it should be clarified explicitely.

55) Figure 6: the difference between the green and light green curves is hard to see. I recommend a better choice of colours. There is a mismatch between the vertical smoothing windows given in the text and those in the caption

56) Figure 7: The large MBL depolarization ratio near the coast is a very interesting features that could deserve more investigation. The figure could benefit from using longitude on the x-axis, instead of time.

---

## Referee Comment (RC2) · Anonymous Referee #2 · 30 Mar 2017

The authors analyzed 4-weeks of lidar observations onboard of a scientific cruise to study Saharan dust transport across the Atlantic. The manuscript is extremely well written, and I have only a minor comments on the writing. I enjoyed reading the manuscript; particularly because of the nice discussion/review the authors did about previous papers on the subject. Methods applied are mostly well justified and valid, and I only have a few points I would like to discuss below. The annotated PDF attached to this response, hopefully, will help the authors to improve the manuscript.

1) The authors talked a lot about the MBL and MAL, and how marine aerosols intrude

into the SAL, but nothing is said about how they actually measured the height of the boundary layer.

2) Most of the trajectories they showed go over biomass burning regions before they actually pass over the desert, where dust would come from. They claim that the analysis showed that marine+dust prevailed (page 6, L 5) but this is definitely not clear from their results alone. It would be necessary to give further evidence for that or point the reader to the figures in other papers that show this is the case.

3) My last point is about the vertical downward mixing mechanism that the authors propose on page 10, L-10-20. I do not think that they can make this conclusion by looking at the trajectories alone. Particularly, if they only compared 2 trajectories, 250m higher and lower than the boundary of SAL. The wind shear could just happen to be a couple of tenths or hundreds of meters higher or lower for that location and time in the GDAS data. One should keep in mind that we are talking about a reanalysis with 50km horizontal resolution over a region where there is no radiosondes or surface meteorology to be assimilated (middle of the ocean). They should compare their own radiosondes against GDAS in the first place to prove that you could use hysplit to distinguish between the trajectories separated by just +-250m. Moreover, for all the hysplit analysis, they could have run hysplit into the ensemble mode, or even in the dispersion mode, so that they would have a much better idea of the probability that the trajectories are point (or not) into the right direction (because they would look into density maps of trajectories, instead of single realisations). This is important as they are following the trajectories for more than 5 days, so the uncertainty is huge.

Please also note the supplement to this comment:
http://www.atmos-chem-phys-discuss.net/acp-2017-130/acp-2017-130-RC2-supplement.pdf

---

## Referee Comment (RC3) · Anonymous Referee #3 · 11 Apr 2017

General comments

The authors report shipborne lidar measurements of aerosol over the Atlantic, a valuable dataset contributing to our knowledge of aerosol properties in this region downwind of the Sahara. From this dataset they are able to infer Saharan dust and marine aerosol properties (e.g. extinction, lidar ratio, depolarization ratio, and Ångström exponent) and loading. Using this information they also explore further the properties of the central Atlantic atmospheric structure, in terms of the marine boundary layer (MBL), the marine air layer (MAL), and the Saharan air layer (SAL). This is an update to work

presented by Kanitz et al. (2014, GRL), a paper written by broadly the same team of authors, and which is well referenced here. The new paper uses the same core dataset as was used and described in the previous paper, and so of necessity there is a certain degree of repetition here.

Given that quite similar work has been presented before, it is important to note what is new here. Section 3 is an expansion of the Kanitz paper, exploring the dataset in more detail beyond what was published in that paper, but starting from the same basic information. Figures 2 and 3 appeared in that paper in a slightly different format, as did half of Figure 6. Figures 4, 5, 7 and the other half of Figure 6 are new. Figure 7 is quite an effective summary of the lidar measurements, meanwhile Figure 4 explores the vertical structure of the lidar signal and depolarization for selected case studies in a more time-resolved manner. The inclusion of HYSPLIT trajectories in Figures 5 and 10 is a useful aid to understanding the possible origins of the aerosols being measured.

Section 4 is more distinct, categorising the atmospheric structure (i.e. MBL/MAL/SAL) using the lidar observations in conjunction with a conceptual model. It is this usage of lidar measurements to inform our knowledge not just of the aerosol over the Atlantic but also of the atmospheric layering that is the newest feature of this paper.

Specific comments

p. 5, line 31: perhaps it would be worth summarising the reasons for these choices of days as case studies, perhaps here or in a table? The reasoning behind these choices is scattered in the text, or left implicit, so it seems to me that for clarity it would be best to make this explicit at an early stage.

Figure 7(e): how do the AODs derived from the lidar measurements compare with the AERONET measurements? The reader can do a visual comparison between this plot and Figure 3, but it would be useful for reference to have some quantitative information on this.

Figure 8: I am not sure that this adds all that much to the discussion within the paper, and indeed it is only referred to very briefly in the text on p. 9, lines 26-27. This information is mostly summarised in Figure 9, perhaps instead the arrows from Figure 8 could be superimposed onto Figure 9? Otherwise I would suggest just removing Figure 8.

Figure 9: would it make sense to reverse the order of the days here? For all of the other plots the time axis went from left to right across the page with the progression of time. This also helped intuitively since the ship was itself progressing from west to east. It would also help with representing the schematic information currently contained in Figure 8, information that is quantified in Figure 9.
* * *

---

## Editor Comment (EC1) · CL Ryder (Editor) · 26 Apr 2017

Please provide responses to reviewers and a revised manuscript showing alterations in the standard way as for ACP/AMT. In your responses, please pay particular attention to the reviewers' comments regarding similarities to the Kanitz et al. 2014 paper, and scientific advances presented in this work compared to what is already published.

---

## Author Comment (AC1)

Dear Editor, Dear Reviewers!

Thank you for careful reading and so many good suggestions. We will provide our answers below, step by step. We completely re-wrote the paper and updated the analysis of our lidar observations taking all your critical comments into account!

The revised version of the paper can be found in this letter below, after the step-by-step answers to the comments of the reviewers.

Because of the many changes, it makes no sense to us to indicate all changes in red, bold, or italic. However, we will carefully respond to all your points.

Let us begin with an overview of the main changes (1-9):

(1) The title is new and similar to the title of the second paper. Part 2 is already published as ACPD version.

Part 1…… (this paper)

**Profiling of Saharan dust from the Caribbean to West Africa, Part 1: Layering structures and optical properties from shipborne polarization/Raman lidar observations**

**Rittmeister,** Ansmann, Engelmann, Skupin, Baars, Kanitz, Kinne

Part 2……

**Profiling of Saharan dust from the Caribbean to West Africa, Part 2: Shipborne lidar measurements versus forecasts**

**Ansmann,** Rittmeister, Engelmann, Basart, Benedetti, Spyrou, Skupin, Baars, Seifert, Senf, Kanitz

Main content of Part 2: Comparison of our four cases 1-4 (presented in detail in Part 1) with simulations (SKIRON, MACC/CAMS, NMMB/BSC-Dust).

(2) The abstract and the conclusions are rewritten to fully cover the paper contents.

(3) In the Introduction, we clearly state and contrast… what is new compared to the first paper (Kanitz et al., 2014).

(4) We introduce the conceptual model already in the Introduction now (Sect. 1.1), and discuss our observations in the context with the conceptual model much earlier and this in two parts (Sect. 3.2.1 and Sect. 3.3.1), we enlarged the discussion…, and we discuss new aspects.

(5) We changed the order of contents in the result sections and therefore also the order of figures, we removed several figures.

Old (submitted) version:

3.1: Cruise overview including the discussion of aerosol layering based on cases 1-4,

3.2 Optical properties of dust and marine particles,

4. Conceptual model vs lidar observations.

New (revised) version:

3.1 Cruise overview,

3.2 Case studies of aerosol layering,

3.2.1 Conceptual model vs lidar observations (part 1),

3.3 SAL and MAL optical properties,

3.3.1 Conceptual model vs lidar observations (part 2).

(6) Figure 1 now includes all figures we use from the the Kanitz (2014) paper. No longer three figures are presented as in the submitted version. In this way, we want to avoid the impression that our paper is just the long version of Kanitz et al. (2014). But we need Figure 1 as an introduction so that the reader does not need the Kanitz paper to get a full overview of the results now available and presented here.

(7) We therefore re-arranged most figures and reduced the number of figures significantly (from 10 to 6).

(8) The summary figure (layer mean optical properties in Figure 6) now includes much more cases. We analyzed all Raman night time observations with cloud-free periods (20 night sessions within the 1-23 May period). Before we presented only 14 night time observations (only those cases which were easy to analyze and to interpret). These 14 cases were most easy to analyze.

(9) We carefully re-checked all points of the discussion in the result section, guided by the comments of the reviewers. We tried to avoid speculations as much as possible.

**Anonymous Referee #1**

…. The main datasets and results are published in a GRL paper, together with some elements of meteorological interpretation with COSMO-MUSCAT and HYSPLIT (Kanitz et al, 2014). That paper highlighted the properties of fresh and aged dust, and characterised it in terms of Angstrom exponent, lidar ratio, particle depolarization ratio, aerosol optical depth, and vertical distribution.

The paper by Rittmeister et al builds on that dataset to conduct further analysis of the results. In particular, the optical properties of the marine aerosol are briefly investigated in addition to dust, and four vertical profiles distributed along the ship's track are investigated in more detail in terms of structure, temporal evolution, and air mass trajectories. A conceptual model is discussed, and a nice sketch of the atmospheric layering across the Atlantic is given (Figure 9). A further follow-on paper is announced within the article text, where more detailed derivations will be shown, including the mass concentrations of fine and coarse dust, and their comparison with aerosol transport simulations.

I have a few concerns with the paper. There is a very large degree of overlap with the previous paper by Kanitz et al, with several figures in common as well as repeated information given in the text, and I think that the paper would really benefit from high lighting new findings rather than going again through material already covered in that paper. Moreover, I believe that the presentation of the material would benefit from a major rewrite. I find that the structure of the paper is not always optimal and that some statements are given as granted whereas a justification or a reasoning could be highlighted. Finally, as a follow-on paper is foreseen, I would recommend submitting those two at the same time and with an organic plan. …. .

… I believe that it will benefit from a careful revision and restructuring, and I will be happy to review it again, if it is resubmitted along the lines that I suggest.

*First of all, we want to thank the reviewer for so many fruitful comments. We see how much time it took to 'analyze' everything in so much detail and then to write all this down. The review helped us a lot. Thank you very much!*

*We got the main message of the reviewer, and this motivated us to re-write the paper as a*

*whole. Before answering all the statements step by step, we want to 'defend ourselves' and want to mention that the Kanitz paper was just a 'quick shot' (a first short introduction based on preliminary analysis of the lidar observations). And therefore we sent it to GRL (and not to JGR as a much larger paper with deep discussions). So, after this 'rapid communication' of Kanitz et al (2014), we now present the final results based on quality-assured FINAL lidar data sets in form of two 'extended' papers. But, sure, we should avoid to have too much overlap with the Kanitz paper and to avoid the impression that we just repeat the contents of the Kanitz paper in a more extended way. But we need the freedom to present a well-balanced discussion in a stand-alone paper, even if there paragraphs in close link to the Kanitz contents. The revised version may be a compromise of all this and also includes the recommendations of the other two reviewers.*

MAJOR POINTS:

1) Title: This title does not highlight what is new in this paper compared to the previous paper by Kanitz et al. Moreover, if the plan is to have two papers I would encourage submitting them as Part 1 and Part 2 more or less at the same time.

*We changed the title to better describe the topics and contents of the paper. We now have part 1 and part 2 (see the titles above).*

2) Abstract: As for the title, the abstract in the current form does not highlight what is new in this paper, and describes findings which are similar to those of the Kanitz et al paper. The first four lines drive the reader to believe that this is the first description of this transatlantic lidar transect, whereas this is untrue. Moreover, the findings that are reported in the abstract (lidar ratio and depolarization ratio) add very little. Both title and abstract should focus on new findings.

*We re-wrote the abstract accordingly.*

3) Several figures are simply reproductions of figures in Kanitz et al (2014). I suggest to omit them here, and write the paper around the new findings instead. In the specific: Figure 1 corresponds to Figure 1 of Kanitz et al; Figure 2 corresponds to Figure 2b+c of Kanitz; Figure 3 corresponds to Figure 2a of Kanitz. Moreover, Figure 6 first and last rows are very similar (although not identical) to the data in Figure 3 of Kanitz et al.

*We now have Figure 1 which summarized all Kanitz results. We definitely need this Fig. 1 in this main R/V Meteor lidar paper. The paper must give full overview of all the activties and it must be understandable without the need of foregoing papers. So we want to show the results in Figure 1, and briefly discuss them. By the way, the map we present in our paper shows more details than the map in the Kanitz paper, e.g., it indicates the SAMUM-1 and 2 and the SALTRACE field sites.*

4) Many findings are presented as new findings and discussed at length, whereas they are instead previous findings from Kanitz et al. Section 2 also does not describe much new material compared to that paper. I would replace the current section 2 with a short summary of the findings by Kanitz et al (not longer than the abstract of that paper). This can then also permit to reduce some of the reproductions of those findings in the current version of section 3.

*To repeat: This is the main paper of the project, and not a minor extension of the Kanitz paper. Therefore: No, we did not follow the reviewer here! To have a stand-alone paper, we need to provide a brief overview of the field activities and observations (see Sect. 3.1).*

5) P3 L30: It is true that a shipborne East-West lidar study has never been performed before 2013, but this has been reported before. This is therefore not the highlight of the present paper. I suggest instead to use an approach like the one on P5 L19: "Kanitz et al (2014) already provided an overview and first results".

*Yes we agree and changed the text in the introduction accordingly (see Sect. 1, page 3, para 2).*

6) P5 L21-22: This interpretation of the lidar data to indicate a MBL and a MAL comes as a surprise here as no reasoning behind it is given. With some experience of lidar signals I can easily recognise where the SAL and the MBL are (with help of the depo- larization plot) but you cannot assume that all readers have this knowledge. The MAL is an new concept to me, I fail to see it at a glance in this plot, and I have not found a convincing explanation in the paper of how the data presented prove its existence.

*Many tropical meteorologists divide the MAL into the sub-cloud layer and the cloud layer. We divide the MAL now in MBL (convective part of the MAL) and the rest up to the trade wind*

*We now give clear definitions of both layers, MBL and MAL in Sect. 1.1 (page 4) and Sect. 3.2, page 8..*

7) There are other interpretations of the data given with little explanation in the text; some of them are listed below (e.g. P5 L29, P6 L8-9, P6 L12-13, P6 L25 and L26, P7

L5-10, P7 L19, P7 L30, P8 L7-8, P8 L34, P9 L3-5, etc.).

*We carefully checked them all (see all our answers below)…*

8) The "conceptual model" presented in section 4 (P9 L12-32) should really be explained in the introduction.

*We introduced Sect. 1.1 in the introduction section to highlight the importance of the model. We tried to integrate the model description in the main body of the Introduction, but that did not really fit…, so we introduced the Subsect. 1.1.*

9) Conclusions: Only 10 lines of conclusions for this work? This is the most important part of the paper. Here you could tie your results with previous research (which you already indicated in the introduction), discuss the caveats and implications, suggest further research, etc. Here is where you justify the benefits of this research in a wider scientific context.

*We extended the conclusion section and now present the main conclusions of the lidar observations shown and discussed before. On the other hand, a conclusion section over several pages would to our opinion be too long, we avoid that…*

OTHER IMPORTANT POINTS TO CONSIDER:

10) P1 L19: dust as "surface-near" plumes because it is part of the BL: please note that the Saharan BL can be extremely deep (up to 5-6 km in a summertime afternoon); therefore this description is inaccurate.

*We removed 'surface-near', yes PBL heights up to 5-6 km were also measured by us during SAMUM 1 in Morocco, so we know that.*

11) P1 L21: I would think that dust lifted to tropopause height is not as common as the paper describes it; most frequently dust plumes are encountered between the surface and 5-6 km.

*This is true over the tropics (including the tropical Atlantic). But our observations over Cyprus and over Tajikistan during the last years tell us another story. As soon as frontal systems and large scale lifting come into play the dust is easily lifted up to 10 km and even higher*

*12)* P2 L1: why do you describe dust as "omnipresent"? Although dust is abundant, I would not think that it is found everywhere (indeed most atmospheric layers around the globe are dust-free).

*We removed this … 'omnipresent'…, and we removed the first sentence of the introduction,*

*concentrate now on long-range transport and dust relevance  for environment and climate..*

13) P2 L21: the distance from the W coast of Africa to the Caribbean is about 6000 km; therefore 10,000 km sounds a bit large.  The same observation applies to P3 L3 where a distance of 5-8,000 km is indicated.

*We need to add at least 1000-3000 km potential transport and dust uptake length over the African continent, and the distance from the west coast of Africa to Barbados is 4500 km and the main Caribbean 5000 km,  so all in all 5000-8000 km  should be ok. We write in the Introduction, now simply…..: more than 5000 km from the main dust sources…. (Sect. 1, page 3, para 1).*

14) P2 L27-30: the change of topic from dust to smoke is a bit sudden at this point in the paper.

*This point is now better introduced in the Introduction, p2, para 4. But smoke is an important aerosol component as the updated discussion (triggered by the reviewers) will indicate later on in the paper, in Sect. 3.3.*

15) P4 L15: between "are available" and "the marine boundary layer" you could add

"from 250 m (full overlap) to XXXX m (limited by SNR), covering ..."

*Done!*

16) P4 L24: you mention separating dust from other aerosol based on the particle de-polarization ratio (I suppose this is the method by Tesche et al, 2009) and you base this on an assumption that the depolarization ratio is 0.3 for pure dust.  I would challenge this, as ageing along the Atlantic path could change the depolarization ratio of the dust component (and you confirm this fact in this paper actually).  I would therefore recommend to take ageing into account when applying this method.

*A new paragraph (Sect. 2.2, page 6, last paragraph) is introduced on the aging aspect. This is need to make generally sure that there is no aging effect (i.e., a decrease of the depolarization ratio with travel time).*

*One essential finding  of all the SALTRACE studies (aircraft, lidar)  including further studies in the Caribbean (Denjean et al, 2015) show that dust aging does not play a role. However, size-dependent gravitational settling, … when the coarse particles are more rapidly removed than the small ones, may cause a decrease of the overall dust depolarization ratio. This minor decrease of the depolarization ratio may be  visible in our SALTRACE lidar observations in Barbados (Haarig et al., 2017a). We discuss all this now in Sect. 2.2, page 6.*

17) P5 L8: I suggest to specify that the radiosondes were launched from the same ship.

*Done!*

18) P5 L13: I recommend to use a word like "prediction" or "computation",  because

"tracking" usually refers to remote sensing observations (radar, satellite, etc.)

*Yes, we changed that accordingly*

19) P5 L15: I recommend to say "used in conjunction" rather than "combined", unless a new aggregate product has been designed that combines both.

*Done!*

20) P5 L29: The colour scale in the plot indicates the magnitude of the range corrected signal, and NOT the different layers.  The attribution of the SAL to "green and yellow colours" (i.e.

range corrected signal between 3 and 4.5) is an interpretation, and as such I think it deserves an explicit explanation in the text.

*Yes, we follow the suggestion, and better leave out to use qualitative explanations by using the colors, only in the case of the marine aerosol layer which is so nicely given in deep blue (in the panels with the volume depol ratio)…, we make an exception (Figure 2, caption).*

21) P6 L8-9: This statement is substantially correct, but it is not formulated in a useful manner; it has to be clear that it is our interpretation that an AOT of 0.05 corresponds to a dust-free pure marine condition, and that 0.7 corresponds to a major dust out- break. The lidar data, the backtrajectories, and correct wording can help support this statement.

*We changed the text accordingly, and we are more careful with wording (see, e.g., Sect. 3.1, page 7, para 2 and 3).*

22) P6 L10: AOT is up to 0.7 (not 0.3)

*Improved*

23) P6 L12-13: larger Angstrom exponent is indicative of smaller particles (and viceversa), not of a given aerosol type. The suggestion that this indicates sea salt or dust is an interpretation, and should be presented as such. In particular, it is reasonable that as dust travels away from source (as presented in this paper), larger particles undergo deposition and therefore the Angstrom exponent increases but the aerosol type remains "dust". This needs probably to be clarified and accounted for.

*Yes, the Angstrom exponent is related to the size distribution, we know. Nevertheless, AE of 0.3-0.7 are indicative for marine conditions in clean marine environments, and values around 0.1 for dust outbreaks. This can be seen from long term AERONET data sets of Barbados (2007 to 2014). We improved wording in the entire Sect. 3.*

24) P6 L14: I appreciate the effort in rationalising what is observed, but before calling the four lidar sections "key stages" I believe that some explanation and discussion could be useful.

*We explain, how we selected these four cases Sect 3.2, page 8, para 1: we took the last (case 4) and the first good dust obs. (case 1), already selected by Kanitz et al. (2014), and then two further cases with 'linear' distance in travel time across the Ocean (case 1: 1 day, case 2: 3 days, case 3: 5 days…). And because the features show the expect trend in full agreement with the conceptual model, we believe they indeed show key stages of the SAL development…*

25) P6 L16-18: It may be worth specifying that this is deduced from backtrajectories. These trajectories pass over hot spots, and therefore are not capable of ruling out a biomass burning component: this could be explicitly discussed.

*This comment forced us to re-think the mixing of aerosols in the SAL. And we checked all lidar observations again whether they are consistent with the assumption that dust and smoke are the main aerosol components in the lofetd SAL. And we now conclude…, there was smoke, and this smoke probably aged during the long range travel as described by Mueller et al. (2007). This effect changes the extinction and backscattering properties and the respective extinction to backscatter ratio from 60sr (for fresh smoke) to values around 30 sr, almost the same as for marine particles (20 sr) which are still larger than the grown smoke particles. However, we also mention that the presence of marine particles in the SAL is also possible… although there is this thermodynamically sharp boundary between MAL and SAL. So, all in all, we try to explain our findings, without giving the feeling that we just speculate… The good news is that dust clearly dominated in the SAL, and that the additional aerosol contribution plays only a minor role except in case 2 where 50% of the dust extinction coefficient is probably caused by smoke. The rewritten discussion on the dust smoke mixture is given in Sect 3.3, page 10 and the following pages in Sect. 3.*

26) P6 L20: To give dust an age (7-9 days), how do you determine at which point along the trajectory it was lifted?

*We discuss that more carefully. We do no longer speculate about the emission day, we leave that open, we only provide the days above the Atlantic, Sect. 3.2, page 8, para 1.*

27) P6 L25: It is unclear how the statement about mass concentrations is justified.

*We removed this statement, it is not needed.*

28) P6 L26: How is the MBL top identified from Figure 4 and how is it found different from the dust base height?

*We computed it from the slope of the range-corrected signal (gradient method, Baars et al., 2008), see Sect. 3.2, page 8, para 3. The difference between MBL top and SAL bases is explained in Sect. 3.2, too. MAL top is equal to SAL base.*

29) P6 L31: smoothing window (365 to 458 m): this is in contrast to the figure, where 457.5 and 562.5 are indicated.

*We corrected that. The smoothing lengths in the figure caption are correct.*

30) P7 L3-4: as Rittmeister et al (2017) is not yet published, may I suggest to cite other existing references about the conversion of optical properties to dust mass concentrations? See e.g.

http://onlinelibrary.wiley.com/doi/10.1029/2007JD009551/full
http://onlinelibrary.wiley.com/doi/10.1029/2000JD900319/pdf
http://onlinelibrary.wiley.com/doi/10.1002/qj.777/full

*We checked the papers and now provide also the reference to Osborne et al., 2008 regarding conversion factors for freshly emitted dust with very pronounced coarse mode fraction, Sect. 3.2, page 10, last para.*

31) P7 L5-10: which analysis showed that smoke does not dominate this airmass? This is not presented at all in this paper. It is definitely not obvious why these fires do not contribute.

*This has now completely changed. The smoke is in…. However, it took us a while before we understood the low lidar ratios in the aged dust plumes 3300 (case 3) and 4300 km (case 4) west of Africa. We were thinking this must be the impact of marine particles, no other aerosol type can explain these obviously low lidar ratios for the non dust component of 30 sr). Only with the paper of Mueller et al. (2007) on aging and growing smoke particles these low lidar ratio can be explained by smoke particles which grew by gas-to-particle conversion, condensation of organic vapors on the particles, coagulation, water uptake, etc. and are thus large and spherical (see Sect. 3.3, page 11). The same we already observed in aged SAL layers over Amazonia during the wet season (Feb – May 2008, Ansmann et al., 2009).*

32) P7 L12-13: you say that the backscatter wavelength dependence is due to long- range transport. However, in Figure 6 this applies also to case 1.

*We skipped …. long range transport.*

33) P7 L19: whereas it is reasonable to think that large particles fall out during long range transport, how does the data support this strong statement?

*We removed this fall-out statement (Sect. 3.3, page 12, para 2).*

34) P7 L30: besides the potential mixing with marine particles, the decrease of depo-larization ratio could also be ascribed to the ageing of dust (removal of larger particles; coating with water and/or other species, etc.)

*Yes! We agree and changed the discussion towards smoke impact in Sect. 3.3, page 10 and the following pages.*

35) P8 L1-2: I would remove the hard numbers here and limit to saying that larger/smaller depolarization ratios are expected.

*No! We need these numbers of depolarization ratios for fine dust and coarse dust (especially in part 2). But we give ranges of values, 0.14-0.18 (fine) and 0.35-0.39 (coarse). This is better than to provide fixed values… , we do it already in Sect. 2.2, page 6…. and later on again in Sect 3.3.1, page 13.*

36) P8 L4: In my opinion, the intrusion of marine particles in the SAL has not really been demonstrated in this paper.

*We removed this statement on dry marine particles in the SAL… It is not needed. However, we mention several times that we cannot fully excluded that marine particles are in the SAL (caused by sea breeze effects when the air masses crossed the coast of West Africa… and later on … caused by strong cumulus convection..) , Sect. 3.3, page 11, para 3.*

37) P8 L7-8: To say that the radiosonde data are in agreement with the lidar observations is again to skip a logical step. Whereas it is clear to me what the authors want to say, I would not think that it is correctly formulated, and as such other readers may find this difficult to understand. I think that the correct statement should be that radiosonde profiles show a consistent layering of the atmosphere with the lidar dataset.

*We agree and changed it, Sect. 3.3, page 12, para 2.*

38) P8 L10-11: I am not sure I understand this. In Figure 6, we see that the RH is large below the SAL base and is small above the SAL base (if we take the depolarization profile as indication of where the SAL boundary is). In P10 L4-5 you clearly acknowledge the sharp change in depolarization ratio at the SAL base: this should be evidence against these vertical exchange processes.

*Yes, we agree, there is this sharp lower edge of the SAL, prohibiting almost any vertical exchange between MAL and SAL. We skipped therefore the discussion on the water vapor profiles in the upper part of the MAL.*

39) P8 L14-16: Omit.

*Done!*

40) P8 L18: Here you mention 16 analysed cases. These come as a surprise because they were never mentioned earlier in the paper.

*This is now mentioned in the Introduction, p3. We extended the analysis and now can present 20 nights with 22 observations in the final figure 6.*

41) P8 L26: Cite literature on the marine LR around 20 sr. Many references exist on lidar ratio of different aerosol types.

For example:

 http://onlinelibrary.wiley.com/doi/10.1029/2006JD008292/full

 http://www.atmos-chem-phys.net/15/3241/2015/
http://www.sciencedirect.com/science/article/pii/S1352231011006108 http://www.atmos-meas-tech.net/6/3281/2013/

*It is not easy to find pure marine conditions and to measure pure marine lidar ratios. Gross et al. (2011) found two days on Cabo Verde (SAMUM 2, winter campaign), and Haarig et al.*

*(2017b) also had a few days at Barbados with pure marine conditions (SALTRACE, winter campaign). During the R/V Meteor cruise we got the largest set of pure marine conditions (almost two weeks continuously). We mention that in the conclusions now.*

*We add Dawson et al. (2015) (CALIPSO observations, page 11, para 3) to the references regarding marine lidar ratios. It is nice to have a global view! But the CALIOP values have to be handled with care. It is an elastic backscatter lidar (many sources of uncertainties….), and combing column extinction with column backscatter leaves space for uncertainties and contributions from the free troposphere…*

42) P8 L32-33: Maybe removing the lower and upper 250 m of the SAL could prevent the fact that smoothing with a ~500 m window introduces information from layers below or above?

*We rephrased the text (Sect 3.3, page 12/13, para 5/1), we simply mention that we take the smoothing lengths into account. But we now clearly state (what we did not do in the submitted version), that the MAL mean values are based on data from 400m height to at least  900m height (when the MBL top is < 900m), and up to MAL top in all other cases…*

43) P8 L34: I think it is really an overstatement to say that a LR of 40 sr "clearly" indicates an impact of marine particles.  The LR of dust is very variable depending of source region (see e.g.  http://onlinelibrary.wiley.com/doi/10.1002/grl.50898/full ). Moreover,  the authors themselves have already acknowledged in this paper that the ageing of dust can reduce its LR.

*Yes! Because of the impact of aged smoke we skipped the …. impact of marine particles on the SAL LR.*

*Regarding the potential LR decrease of aged dust we disagree and improved and strengthened our argumentation at several places in Sect 3.3. All SAMUM and SALTRCE observations point to the fact:  LR for western Saharan dust is 50-60sr! Of course, LR is different for Middle East and Asian dust, and even for eastern Saharan dust  the LR is smaller, …towards 40 sr (or even lower). But for western Saharan dust it is clearly 50 to 60sr.*

44) P9 L1-3:  Again I believe that the comments on the depolarization  ratio are too sharp, and I would moderate them in terms of possibilistic statements.

*Yes, as already mentioned above, now we provide ranges …0.14-0.18 (fine dust) and 0.35-0.39 (coarse dust), Sect. 3.3.1, page 13. These are new aspects (i.e., next steps in the use of depolarization ratios to separate fine and coarse dust as presented in the follow-up article, Ansmann et al., 2017) and summarized in Mamouri and Ansmann (2017).*

45) P9 L3-5:  I believe once again that the authors have no evidence for saying that in proximity of the African continent there is no MBL. Indeed,  models and campaigns indicate that such a layer exists near the coast. A more plausible explanation could be that the large depolarization  ratio is due to fall out of large dust particles from the SAL above.

*We changed the text and skipped this statement. But we were discussing the built up of the marine MBL, and that takes time, when continental air masses travel over the ocean… It takes at least 500 km before a convective marine MBL could develop in the continental air mass over the ocean.*

46) P9 L9-10: This concept has been repeated several times throughout the paper, but I am not persuaded by the arguments as already commented.  Ageing mechanisms are plausible causes. There is also no need to repeat a same concept so many times.

*Yes, we now try to avoid to mention 'the same concept' many times. However, we need a logical structure of argumentation, so if we repeat statements, they are to our opinion necessary.*

*Regarding aging! .... we mention aging now, but with respect to smoke, because in this case, aging effects are really visible in the observations (Sect. 3.3, page 11, para 2).*

*Regarding dust aging effects! If there would be clear hinds in the literature that aging effects have an impact we would integrate that into our discussion. But we do not see them in the literature, and other scientists do not see them… Yes, aging may be an issue, atmospheric chemists 'attack' us (lidar people) with this argument since 10 years, but obviously these aging effects are not able to change the optical properties of dust significantly (especially dust-particle-shape-related optical properties remain unchanged).*

*To continue… All the SAMUM and SALTRACE aircraft and lidar observations tell us that there are no significant changes in the dust properties from Africa to the Caribbean. And although all our results are consistent, we shall still argue, there must be an aging effect? The main aim of the SAMUM and SALTRACE campaigns was to find these changes, but the main result is: We did not find these changes!*

47) P9 L19: unclear: "except in disturbance".

*We skipped it!*

48) P10 L14: smoke? not discussed much in this paper

*Yes, but now the discussion has changed significantly.*

49) Figure 1: omit figure as it is part of Kanitz et al. Continents are not clearly visible.

*The reviewer is right, we gave the Kanitz 2014 result too much space in the submitted version. We reduced that a lot. However, we need the new Figure 1 (as an introductory figure) in which we collected several Kanitz 2014 results. The map is updated (continents may be a bit too dark, but scientists should know where Africa and Caribbean is…), the center panel shows an overview of all lidar observations … this is good in connection with the final figure 6, in which we summarize the results of 20 nights out of 22 nights, … and the bottom panel shows the AOT and Angstrom exponents which we need in our discussion in Sect. 3.1, and also later in Sect. 3.3.*

50) Figure 2: omit figure as it is part of Kanitz et al. The caption does not describe the figure, instead it tries to interpret it.

*We leave the range-corrected-signal color plot in … as the central plot of the new Figure 1, see reasons above.*

*We disagree (…concerning 'interpretation')! We lidar people always tend to say first what parameter is plotted instead of…. What is shown, why do we show this figure? So, we re-phrased our figure captions but want to follow our own 'philosophy' how to present the necessary information in the captions.*

51) Figure 3: omit figure as it is part of Kanitz et al. The gray-shaded areas are hardly visible when this is printed. It is unclear what criteria where used to delimit them. A longitude x-axis would probably be more useful than a time axis.

*We leave the figure in and did not change it. Yes, dust periods seem to be not well defined…, and the yellow lines and grey areas may be not very well visible but this is just background information and therefore we selected some kind of background colors. The gray-shaded areas show days where we detected dust.*

52) Figures 4-6 and 9-10: I suggest reversing the order of cases 1-4, to reflect the order of the discussion in the text (P6 L16-22). This would also have the benefit to have the Easternmost panel on the right (and the Westernmost on the left) in Figure 9, i.e. like one would see it on a map.

*We had this order 4,3,2,1 in the very beginning (when writing this paper). Then the co-authors told us, please change that. So, now we do not want to change that again. And west-to-east arrangements (from left to right) and case 1 to case 4 from top to bottom is ok, we think. So, we changed the sketch (now Figure 3) to better illustrate the west-to-east aspect.*

53) Figure 4: Caption does not explain what is shown (RCS and VDR), does not clarify how the MBL and MAL are distinguished. The data within the incomplete overlap should be treated as missing data instead of commenting on the "blue area" at the bottom. An indication of longitude for each case would be useful. Blue areas in the right hand panel indicate low VDR, which is indicative of dust-free layers; they do not directly indicate dust-free layers.

*Figure 4 is now Figure 2. We 'optimized' the caption, but followed out 'caption philosophy' mentioned above: We begin with something like a head line indicating: what we want to show…: Here, the layering structures! And then we tell the reader, what parameters are plotted, with what resolution, and then write, what details can be seen. We hope the caption text is acceptable.*

54) Figure 5 caption: why do you say that the 500 m level is always within the MBL and the arrival heights 1500-3000 m are always in the SAL? I suppose this is indicated by the lidar profiles, but if it is the case it should be clarified explicitly.

*Figure 5 is still Figure 5 (in the revised version). Figure 5 comes now almost at the end, and at to that time the reader should be familiar with MAL and SAL, and does not need more information than that. Nevertheless, we improved the caption a bit, rearranged the text.*

55) Figure 6: the difference between the green and light green curves is hard to see. I recommend a better choice of colours. There is a mismatch between the vertical smoothing windows given in the text and those in the caption

*Figure 6 is now Figure 4. Yes we agree, the colours light green and green are not easy to distinguish. But this not so important to see the differences. All these green curves show results for 532 nm. The figure is already very busy with all information. So, we try to distinguish clearly the wavelengths (blue and green), and leave aspects like near-range and far-range data analysis in the background (light green and green).*

56) Figure 7: The large MBL depolarization ratio near the coast is a very interesting features that could deserve more investigation. The figure could benefit from using longitude on the x-axis, instead of time.

*Figure 7 is now Figure 6. Yes, we agree with the reviewer regarding the MBL depol ratio near the coast (20-23 May). In the improved Figure 6, the dust impact on the depol ratio and AOT is clearly visible. We discuss it in Sect. 3.3, pages 12+13.*

*To better see the links between all these longitudes, latitudes, dates and time, and case numbers, we have Figure 1… Therefore, we did not change it.*

*We provide more precisely how we calculated the MAL values (if the MAL top is below 900 m) we always calculated the MAL mean value from the values from 400-900m, disregarding the fact that the MBL was … maybe … at 600 m. Now we integrated more cases, which were not given in the submitted version. And especially for the last days 20-23 May.*

*Triggered by this valuable comment, we analyzed the full data set again and filled many empty spaces. In the first version, we just selected what the first author Franziska Rittmeister analyzed. And she hesitated to show too many 'crucial' cases with not well defined MBL and MAL and all this in the near-range of the lidar … Complicated and crucial cases are now added.*

…. I only have a few points I would like to discuss below. The annotated PDF attached to this response, hopefully, will help the authors to improve the manuscript.

1) The authors talked a lot about the MBL and MAL, and how marine aerosols intrude into the SAL, but nothing is said about how they actually measured the height of the boundary layer.

*We now provide clear definitions of MBL (convection part of the MAL) and the MAL (layer up to which the depolarization ratio is low) in Sect 3.2, page 8. We already introduce the MAL and MBL layer discrimination in the Introduction (Sect.1.1, conceptual model description)*

2) Most of the trajectories they showed go over biomass burning regions before they actually pass over the desert, where dust would come from. They claim that the analysis showed that marine+dust prevailed (page 6, L 5) but this is definitely not clear from their results alone. It would be necessary to give further evidence for that or point the reader to the figures in other papers that show this is the case.

*The comments of reviewer 1 are similar. So we were forced to re-think and re-analyze the results, and we now come up with an improved discussion and conclusions which provide a more consistent picture of aerosol mixing in the SAL, a better and more reasonable agreement with the fire maps and backward trajectories (see Sect. 3.2, pages 10 and 11). Yes, there was smoke in the SAL, in terms of extinction of 10-20% (cases 1,3,4) and up to 50% in case 2 (14 May 2013 observations). We discuss that in Sect. 3.4. When writing the submitted version, we were confused by the fact that the lidar ratios for case 3 and 4 were so low which we could only interpret as marine aerosol contribution. This option is still mentioned. However, after checking the literature again (especially our own papers! Mueller et al., 2007, on growing smoke particle during long range transport by gas-to-particle conversion, condensation, coagulation, water uptake… ), we found out that the non-dust component in cases 3 and 4 can be aged smoke with changed optical properties (the lidar ratio decreased from 60 sr in case 2, to about 30 sr after 5-10 days of travel). This is in agreement with the literature, we mention all this now. However, we leave the conclusions open. We cannot exclude the possibility that convection (trade wind cumuli evolution) pushes some marine air into the SAL after 3000-4000 km of travel over the ocean, although there is the strong barrier (rather stable stratification) between MAL and SAL.*

3) My last point is about the vertical downward mixing mechanism that the authors propose on page 10, L-10-20. I do not think that they can make this conclusion by looking at the trajectories alone. Particularly, if they only compared 2 trajectories, 250m higher and lower than the boundary of SAL. The wind shear could just happen to be a couple of tenths or hundreds of meters higher or lower for that location and time in the GDAS data. One should keep in mind that we are talking about a reanalysis with 50km horizontal resolution over a region where there is no radiosondes or surface meteorology to be assimilated (middle of the ocean). They should compare their own radiosondes against GDAS in the first place to prove that you could use hysplit to distinguish between the trajectories separated by just +-250m. Moreover, for all the hysplit analysis, they could have run hysplit into the ensemble mode, or even in the dispersion mode, so that they would have a much better idea of the probability that the trajectories are point (or not) into the right direction (because they would look into density maps of trajectories, instead of single realisations). This is important as they are following the trajectories for more than 5 days, so the uncertainty is huge.

*We skipped the figure with the trajectories and skipped the whole discussion which was too speculative. The surprisingly low depolarization ratio and other optical properties (in Figure 6) which point to almost undisturbed clean marine conditions in the MAL (and to the absence of falling dust particles…) is taken as the main reason that there must be other mechanism active besides gravitational settling which lead this efficient removal of dust from the MAL*

*(such as turbulent downward mixing, cloud and precipitation processes). We do all the discussion carefully.*

*We checked this annotated PDF of the reviewer, and included almost all suggestions into the revised manuscript. Thank you for taking the time to do all this.*

*Some questions came up in the PDF (we provide just the answers):*

*The MBL top height is determined from the range-corrected signal profiles by using the gradient method (Baars, 2008). See Sect. 3.2, page 8.*

*In the former Fig. 2 (now Figure 1, central panel) there is already a white line to indicate the MAL top, the top of this marine aerosol layer is defined by the volume depolarization ratio profile. The height at which the depol ratio jumps from around 0.03 to more than 0.1 is defined as MAL top (or SAL base, se Sect. 3.2, page 8). We do not want to have another line to indicate the MBL (convective zone of the MAL). Maybe we missed the point of the reviewer.*

*Because of the comments of all three reviewers, we re-analyzed the data and re-checked our conclusions. As mentioned: Yes there was smoke in the SAL (about 10% extinction contribution in case 1, 20% in cases 3 and 4, and 50% in case 2). All our measurements (depol ratio, lidar ratio, polarization technique in combination with Raman extinction technique) point to smoke when we assume that smoke is growing during the long-distance transport so that the lidar ratio decreases from about 60 sr for almost dry smoke particles to about 30 sr for aged and water-rich particles. Smoke particle growth was discussed by Mueller et al. (GRL, 2007). However the smoke impact is low, except for case 2. And we state, that we cannot fully rule out that marine particles were also present in the SAL, because of the long transport over the ocean and the probability for cumulus convection and penetration into the lower part of the SAL.*

*We skipped the discussion on enhanced depolarization ratios when marine particles dry. This discussion is needed in part 2, but not in this article, part 1. It would be too confusing if we would discuss all potential impacts, even the minor ones.*

*We re-wrote the abstract and the conclusions (Sect. 4, pages 14+15) which were definitely too short. Now they should cover the entire contents of the paper.*

*Engelmann (2011) and Jaehn (2015) made Doppler lidar observations of vertical winds and heat isalnd simulations, respectively. And the observed data and modeled data show enhanced vertical mixing and thus the chance for dust to become better distributed over the entire MBL. We kept this part short in the text (Sect. 3.2.1, page 9, para 3).*

*We did not check the GDAS data for possible wind shear indications at MAL top. We left it open whether this undercutting effect has a strong or less strong impact. It would remain speculation.*

*Figure 4 (now Figure 2), rainbow colors! Yes we know about the problem that this color scale can produce layers for our eyes although they are physically not given. But, there is no better way to show colorful pictures (this is a psychological aspect, optimistic way of presenting the world). And even CALIPSO uses rainbow colors, and they have a rather tricky scale structures, the scale is not the same for the entire range of values, and this is much more dangerous, we believe. We did not change our way to present color plots, we use this style since more than 25 years now.*

*Below 250 m the signals decrease rapidly (when looking downward towards the ocean surface...) because of the incomplete overlap. However, this effect cancels out for the volume depol ratio which is based on signal ratios. So, we have always an idea about aerosol stratification down to the surface.*

*Figure 5 (backward trajectories and fire spots): As mentioned we change the discussion towards a mixture of dust and smoke (Sect. 3.3, pages 10+11).*

Anonymous Referee #3

General comments

The authors report shipborne lidar measurements of aerosol over the Atlantic, a valuable dataset contributing to our knowledge of aerosol properties in this region down- wind of the Sahara. From this dataset they are able to infer Saharan dust and marine aerosol properties (e.g. extinction, lidar ratio, depolarization ratio, and Ångström exponent) and loading. Using this information they also explore further the properties of the central Atlantic atmospheric structure, in terms of the marine boundary layer (MBL), the marine air layer (MAL), and the Saharan air layer (SAL). This is an update to work presented by Kanitz et al. (2014, GRL), a paper written by broadly the same team of authors, and which is well referenced here. The new paper uses the same core dataset as was used and described in the previous paper, and so of necessity there is a certain degree of repetition here.

*First of all, thank you for the nice and long review!*

*See our description on the differences between the Kanitz 2014 paper and the new Rittmeister 2017 paper (Sect. 1, page 3, paras 2 and 3).*

Given that quite similar work has been presented before, it is important to note what is new here. Section 3 is an expansion of the Kanitz paper, exploring the dataset in more detail beyond what was published in that paper, but starting from the same basic information. Figures 2 and 3 appeared in that paper in a slightly different format, as did half of Figure 6. Figures 4, 5, 7 and the other half of Figure 6 are new. Figure 7 is quite an effective summary of the lidar measurements, meanwhile Figure 4 explores the vertical structure of the lidar signal and depolarization for selected case studies in a more time-resolved manner. The inclusion of HYSPLIT trajectories in Figures 5 and 10 is a useful aid to understanding the possible origins of the aerosols being measured.

*Thank you for helping us to define the differences between the Kanitz 2014 paper and the new one. As mentioned above, we put all the Kanitz plots used in the new manuscript in Figure 1. We need them to have a logical introduction. Figures 4,5,7 in the submitted version are now Figures 2, 5, and 6 in the revised version. The old Figure 6 shows already two of the four case studies. But now we use the quality-assured final lidar data. The differences are not very large, but at least, now we use the best available data. The new Figure 6 (the old Figure 7), now includes 20 nighttime observations (in the older Figure 6, we showed 14 nights only). We omitted Figure 15 (the second Hysplit plot) because the discussion was too speculative, we shipped this discussion so that we also skipped Figure 15.*

*There is a new paragraph in the Introduction (Sect. 1, page 3) in which we summarize to what extend the Kanitz et al. (2014) results are similar and what is new in the new papers (parts 1 and 2).*

Section 4 is more distinct, categorising the atmospheric structure (i.e. MBL/MAL/SAL) using the lidar observations in conjunction with a conceptual model. It is this usage of lidar measurements to inform our knowledge not just of the aerosol over the Atlantic but also of the atmospheric layering that is the newest feature of this paper.

*Yes we agree and changed the title to include the aerosol layering aspects.*

*Please note that we rearranged the paper as described in the beginning of the reply letter. Sect 4 of the submitted version is now split into Sect. 3.2.1 and 3.3.1.*

Specific comments

p. 5, line 31: perhaps it would be worth summarising the reasons for these choices of days as case studies, perhaps here or in a table? The reasoning behind these choices is scattered in the text, or left implicit, so it seems to me that for clarity it would be best to make this

explicit at an early stage.

*We state this now in Sect. 3.2, page 8, para 1. We selected the first and the last dust measurements as cases 4 and 1, and then we selected two further cases so that we have a series of observations of dust after a traveling time of 1 day, 3 days and 5 days over the tropical Atlantic. And when comparing the results with the conceptual model then we found these cases represent different stages of dust layering in agreement with the model.*

Figure 7(e): how do the AODs derived from the lidar measurements compare with the AERONET measurements? The reader can do a visual comparison between this plot and Figure 3, but it would be useful for reference to have some quantitative information on this.

*Figure 7 is now Figure 6. We improved the discussion in Sect. 3.3 (pages 10-14) in general regarding a comparison and consistency analysis of AERONET and lidar-derived AOTs and also Angstrom exponents. But in the old Figure 7 (now Figure 6), we now skipped the marine AOT because we always start at 400m above the lidar and made an estimate for the lowest 400m regarding the AOT below that height. We do not want to show that anymore. We only show the SAL optical depth now. We found good consistency between the sun photometer and the lidar observations.*

Figure 8: I am not sure that this adds all that much to the discussion within the paper, and indeed it is only referred to very briefly in the text on p. 9, lines 26-27. This information is mostly summarised in Figure 9, perhaps instead the arrows from Figure 8 could be superimposed onto Figure 9? Otherwise I would suggest just removing Figure 8.

*Yes, we followed this suggestion and combined Figures 8 and 9 (submitted version), now given as Figure 3.*

Figure 9: would it make sense to reverse the order of the days here? For all of the other plots the time axis went from left to right across the page with the progression of time. This also helped intuitively since the ship was itself progressing from west to east. It would also help with representing the schematic information currently contained in Figure 8, information that is quantified in Figure 9.

*Figure 9 (together with Figure 8) is now Figure 3, and we changed the former Figure 9 accordingly.*

[revised manuscript text omitted]

---

## Author Response (AR2)

Dear Editor, Dear Reviewer!

Thank you again for careful reading and taking the time to think about the paper contents. We enjoyed the suggestions and considered most of them (90%) in the revised version. The revised version of the paper can be found in this letter below, after the step-by-step answers to the comments of the reviewers.

We improved the map in Figure 1 as much as possible, but did not change the other figures (the arguments are given below).

Here are our answers and how we considered all the suggestions:

**GENERAL COMMENTS:**

**The paper is largely improved, showing a better presentation of the material and consequently it is easier to follow than in the first version (the first version seemed to have been assembled a little bit in a rush). The conceptual model being discussed at the beginning, with a clear definition of the layering considered (SAL, MAL, MBL), it is now possible to follow the authors' reasoning. Conclusions have been written (whereas they were almost unexistent in the previous version), although I think that they are not yet fully mature.**

We considered the suggestion to improve the conclusion section.

**As opposed to Kanitz et al, the current paper presents quality assured final data. It would be good to highlight in section 2.2 what different processing and QA tests have been implemented compared to the previous paper.**

We provide now the requested information in Section 2.2 (second paragraph).

**The authors argue strongly against aging effects in the optical properties for the observed dust across the Atlantic, and some of their arguments appear convincing. On the other hand, however, they seem to confirm that the PDR would decrease in the case that the dust downstream were finer than upstream, and that the LR would be modified in the case that the dust were mixed with other species. I believe that there is a contradiction here: mixing with other species and modifications of the PSD are indeed aging mechanisms, which is something that the authors do not seem to recognise. I suggest therefore to reword the paper in such a way as to reflect these considerations.**

We follow the suggestion of the reviewer (see Section 2.2, paragraphs 5-8).

**Whereas PDR and LR can help identify aerosol type, and indeed aerosol typing algorithms are possible based on optical properties, it is not true that these algorithms always give the correct answer. Moreover, the "typical" optical properties for a same aerosol type may vary (e.g. dust from different source regions). I believe therefore that the authors' statements on this matter are too sharp. We should always bear in mind that these are not direct measurements of composition, and e.g. a same set of optical properties measured in a different area of the world (or when we know that different sources are active) could lead to different conclusions. I believe therefore that these statements could be softened, reflecting the fact that the optical measurements give indications, that need to be considered in conjunction with the understanding of sources and transport. The most experienced can understand these nuances even when they are omitted, but our papers are read by our students, and they learn from us. We want our students to develop a critical understanding of aerosol typing, rather than a simplified reality.**

We follow this concept and softened many statements at many places.

**Similarly, sharp statements are made by the authors on the percent dust in a mixture with something else, without much explanation to the reader about how such quantities are determined, or what the underlying assumptions are. This type of quantification comes most probably from assuming a given depolarization for pure dust and a different one for the "other aerosol": these assumptions are fine, but like all assumptions,**

they must discussed and their consequences and uncertainties must be mentioned. One more assumption (too often overlooked, although it is fundamental) for this method to work is that the two aerosols are externally mixed, and this should be clearly stated.

We discuss the assumptions of the polarization lidar technique now in Section 2.2 in very large detail. We give a lot of information regarding a potential impact of internal mixtures (when assuming an external mixture), we include new references here. Nevertheless, at the end we still can conclude that the polarization lidar technique is very robust, the uncertainties are not too large. We think, the paper should not be overloaded with lidar retrieval explanations. We give always clear references that will provide deeper insight.

The examination of the backtrajectories brings the most convincing indication concerning the mixing of dust and smoke in the observed aerosols. What the data available do not allow us to determine, is whether the mixture is internal (e.g. coating or adsorption of smoke on dust particles) or external (co-existing but separate dust and smoke particles). Although it will not be possible to resolve this matter from the data available, I think that it has to be mentioned as an area where further research can bring important insight.

As mentioned, we discuss the aspect of internal mixtures now in large detail…, provide several new references.

Large depolarization ratios for the MAL near the African coast are shown in a figure and mentioned in the text, but they are not discussed although they are a very surprising result. I believe, however, that this may be due to an incorrect separation between the two layers using a fixed altitude set at 900 m. A variable boundary, inferred from the lidar dataset, could be used for a correct representation.

Yes, when using a fixed upper boundary of the data analysis range, there is an impact of dust (close to Africa, at heights < 900 m, 20-23 May). We discuss this point now in more detail. However, we did not change the data analysis, because these effects influence the results for the last days (20-23 May) of the cruise only. We cannot provide any results for the lowest 250-500 m alone. We explain that in much detail in Section 3.3.

It seems from the answers to the previous review that the authors are reluctant to make any graphical modification to figures. Whereas this may represent some additional effort, in some cases it is necessary to ensure readability, and I recommend to update figures when issues are identified.

We checked all figures again, and give our answers below. We improved Figure 1 (top panel) as much as possible. We did not change the other figures, because we do not see problems with the printed versions. We believe the printer of the reviewer causes the problems. Our printers in the institute provide excellent printout quality. All grey areas are clearly visible.

My recommendation is for a major review.

MAJOR POINTS:

1) To completely remove the impression that the current paper is the "long version" of the Kanitz paper, I would be more explicit in the abstract's opening sentence as follows: "We present here final and quality assured results of multiwavelength polarization/Raman lidar observations of the Saharan Air Layer (SAL) over the Atlantic. Observations were performed aboard the German research vessel R/V Meteor during the one-month transatlantic cruise from Guadeloupe to Cabo Verde over 4500 km from 61.5W to 20W (mean latitude ???) in April-May 2013 that was reported in a previous concise paper."

Done!

2) Whereas microphysical properties can be inferred from optical properties (indirect measurement), the type of lidar that the authors use cannot make a DIRECT measurement of microphysical properties. Therefore, omit "microphysical" on P3 L10.

Done!

**3) P3 L34: the SAL is not only a dust layer; it is principally a hot and dry airmass. The way this is presented here seems to indicate that dust is the main property of the SAL, whereas this is incorrect. See also the next comment.**

We changed the text accordingly, re-checked the papers of Karyampudi et al. (1999), of Carlson and Prospero (1972), and Colarco et al. (2003), and rewrote the Section 1.1 (conceptual model) to be very precise.

**4) P4 L11-13: the authors report some geometric information on the SAL, but it is unclear where this comes from. Is this still extracted from Karyampudi et al? Note that this sentence is not very informative when it says that "the SAL top is assumed to lower due to [...] a general lowering of the dust-layer top". The authors also attribute the lowering of the SAL top to the depletion of giant particles, but (1) this seems to be in contradiction with the position that the authors take towards the absence of aging mechanisms for dust (the change in the PSD due to deposition would be an aging mechanism); (2) it is unclear how the sedimentation of the largest particles would affect the SAL layer top (top of the hot and dry layer); and (3) the radiosondes presented in the current paper clearly show that the airmass above the dust layer is different, hence the lowering of the SAL top is due to atmospheric dynamics rather than sedimentation. If sedimentation had been the driver, the air above the dust layer top would conserve its thermodynamic properties (hot and dry). I would assume that travel for a long time over the cold surface of the ocean could progressively reduce convective exchanges in the SAL and hence cause it to become shallower. Divergence of the air flow could potentially also reduce the SAL's vertical extent (airmass redistribution over a wider area and hence thinning of the layer for mass conservation).**

Triggered by these comments, we went through the three above mentioned papers and re-wrote carefully the text in Section 1.1. Now the conceptual model is explained as the authors explain it. …. And we take then the freedom to leave out, what is not just convincing …( to us and to the reviewer). We now say that aging of dust includes cloud processing and chemical again, PSD changes, etc. in Section 2.2.

**5) P4 L14-16: I suppose that these considerations are about the potential of wet removal mechanisms associated with the stratocumulus deck at the top of the MBL: is it please possible to clarify? Clouds have some times also been observed at the top of the SAL: what about their ptential to trigger wet removal mechanisms?**

As mentioned, we clarified these points. Most of the wet deposition occurs within the MAL and is linked to the trade wind cumulus clouds and cloud layers. Deep convection contributes as well sometimes… However, we leave out to discuss cloud layers at the top of the SAL and their wet-deposition impact. We never observed such clouds during the cruise, SAMUM-2, and SALTRACE 2013, 2014. The humidity in the SAL is always quite low and does not allow cloud formation within the SAL.

**6) P6 L4-21: As discussed in the general comments, I find this text to be contradictory. On one hand the authors take position against the existence of dust aging mechanisms, and on the other hand they admit that the PSD may shift towards smaller particles due to removal mechanisms, with a consequent reduction of the PDR. I feel that this is precisely an aging mechanism. Moreover, whereas the authors claim that fine dust's PDR is 0.14-0.18, I do not understand why all the intermediate values between 0.3 and 0.14 are not possible during different stages of the aging. Finally, how can aging due to deposition (PSD change) be distinguished by lidar from aging due to the uptake of water, smoke, or pollution, if the effect on the PDR is similar?**

Keeping this comment into consideration, we changed our argumentation. Aging includes PSD changes now. We changed and adjusted the whole discussion (Section 2.2). However, we observed always total (fine plus coarse) dust depol values around 0.3 when we were sure there is only dust (SALTRACE summers of 2013, 2014, above 2 km height, Haarig et al. 2017a, Gross et al. 2015). There are always coarse dust particles which dominate the overall optical effect. It is impossible to measure particle depolarization ratios of about 0.23 (at

532 nm) in pure dust layers! For the Haarig et al. (2017) paper, we analyzed CALIOP data measured between Africa to Barbados. The depol values are at all close to 0.3, and not continuously distributed between 0.31 and 0.15 (at 532 nm).

**7) P7 L11: As discussed in the general comments, the large PDR of the MBL close to the African coast may possibly be ascribed to a methodological flaw, as the boundary between the two layers is set at a fixed height. Models often show a very shallow MBL height under the dust layer just off the African coast.**

Yes, the reviewer is right. But we have no alternative. We can only start our data analysis at 400 m above the lidar (because of overlap problems), and need at least a 500 m layer to get trustworthy results (only then signal to noise ratios are low enough). We explain that now in detail when we discuss Figure 6 in Section 3. The alternative option would be to leave out to show data. But we do not like that.

**8) P7 L14-16: Soften statements! An AOD alone cannot be used in aerosol typing and other considerations are needed (lidar observations, meteorology, etc.), whereas the authors say sharply that an AOD of 0.05 automatically equates to marine conditions and one of 0.7 to a dust outbreak. L23: "indicates" --> "suggests" L25: "marine" --> "finer"**

We follow the reviewer, are bit more precise now (section 3.1). However, we did not say that an AOD of 0.05 'automatically' indicate marine conditions.

We changed wording (L23, L25).

**9) P8 L27-30: this is plausible, but from the analysis presented in this paper there is no proof of this airmass history. This should be made clear.**

We state this now (at the end of Section 3.2, as suggested by the reviewer).

**10) P9 L5: Unless there exists a specific Meteosat "wet deposition" product (which I am not aware of), I suppose that cloud cover was used as a proxy. Therefore please soften the statement: "potential impact of wet deposition by strong cumulus development" --> "presence of strong cumulus". L6: delete "wet deposition"**

We changed the text accordingly in Section 3.2.1.

**11) P10 L1-7: the statement that trajectories did not cross areas with biomass burning is contradicted by Fig. 5. Also, soften statements on the conclusions drawn from optical properties as follows: (1) specify how the estimated 10% non-dust extinction is computed; (2) Angstrom exponent indicates large particles; hence no external mixture with a non-dust component; but what about an internal mixture? An internal mixture would also display large particles and hence a low Angstrom exponent!**

We follow mostly the suggestions of the reviewer, but do not explain the separation technique, because this is done in part 2 (Ansmann et al., 2017), we provide the reference in this context. In Section 2.2, we provide an extended discussion on external vs internal mixture.

**12) P10 L8: soften statement! "significant" --> "possible"** Done!

**13) P10 L10-11: soften statement by being possibilistic on aerosol type and by explaining how the 50% non-dust is estimated.** Done in Section 3.3.

**14) P10 L15: soften statement! add "potentially" before "contained smoke"** Done!

**15) P10 L20-21: soften statement!**

Done (Section 3.3). But we give numbers., otherwise our efforts to analyze the lidar data quantitatively by means of the powerful polarization technique make no sense. The uncertainties are not that large, as it seems to be the case for the reviewer.

**16) P11 L1-9: these computations are valid for an external mixture. This should be stated. If dust is coated with "something else" then the Tesche et al (2009) method cannot be applied.**

We provide an extended discussion on external vs internal mixture in Section 2.2. That should be sufficient for the entire paper.

**17) P11 L16: soften statement! "as obviously observed with the shipborne lidar" --> "the shipborne lidar observations are compatible with an aged smoke layer". As for above, no consideration for internal mixing is provided: please discuss.**

We state that (… compatible..). But we leave out to mention that we ignore internal mixing effects. This was already  clarified in detail in Section 2.2.

**18) P13 L2: see above (general comments) on using a fixed separation between layers at 900 m.**

We explain why we used the 900 m height level in the data analysis, and that the MAL data from 20-23 May are therefore dust contaminated.

**19) P13 L17: soften statement! "indicates" --> "suggests"; "smoke" --> "smoke or sea salt"   Done!**

**20) P13 L31: soften statements! "leads" --> "should lead"; "causes" --> "should exhibit"   Done!**

**21) P13 L33: "but this is not found". Vertical mixing within the SAL could explain the constant PDR across the layer, and only particles that make it to the bottom of the layer into the MAL would then be removed. Your radiosonde measurements could provide a unique insight on the SAL thermodynamics and ability to mix vertically. And if I read further to P14 L9-15, I see that you also explain this well. Therefore, the feeling that there is no new finding here, but a confirmation of previous findings. The whole way this is presented could be therefore reversed: initially present what other articles say, and then illustrate the position that your observations show. This would highlight that your contribution is to confirm given hypotheses. Note: another conclusion from this is that the next E-W Atlantic transect measurements would benefit from vertically resolved PSD measurements (although technology for this is just beginning to mature).**

We follow the suggestion, but be leave out to mention here (right before the conclusion section) that airborne observations of PSD profiles would be desirable. This is done in the conclusion section.

**22) I feel that the conclusions are not yet mature for publication and I suggest that the authors spend a bit longer time to refine them. P 14 L21: I feel that it is an overstatement that the data available "permitted the study of dust removal aspects in large detail". This should be softened by saying that the data presented are compatible with previous papers (see previous point). L24-25: the paper does not present evidence on dust removal below the SAL. L26: "less efficient removal than expected": this is not correct, as literature that you also cite had already come to the same hypothesis; it is much better to say that your findings support those conclusions. L31: anthropogenic haze has not been discussed in this paper; "was" --> "could be". P15 L1: "was caused" --> "was estimated to be caused". L2: "grew" --> "may have grown". L26: mandatory? who has decided this?**

We agree, and rewrote the entire conclusion section. We extended the conclusions towards the wish for long-distance airborne in situ observations across the Atlantic.

**23) I believe that the main points that the conclusions should focus on are (1) atmospheric layering observed; (2) potential mixing of dust and smoke; (3) similarities and differences of observations from**

**Karyampudi's conceptual model; (4) similarities and differences of observations from different papers on sedimentation; (5) the need of more observations of the SAL over the tropical Atlantic; and (6) what other observations, in addition to the ones you had on the research vessel, would be desirable to answer the most compelling questions.**

We state that now in the conclusions.

**24) Figure 3: the gravitational settling (red arrow) and the exchange mechanisms between layers are not demonstrated in this paper, and the evidence presented by the authors seems to be against them. Therefore they should be omitted from this figure to reinforce the message.**

Here, we do not follow the reviewer, because without these 'symbols' for the main removal processes the sketch becomes almost senseless. That would be a pity. The sketch needs these process-indicating symbols. These are the main downward transport mechanisms in the conceptual model. We make that more clear in the text and the Figure 3 caption, … that these are the main dust removal processes in the conceptual model

**5) Figure 6: I believe that the fixed separation at 900 m between SAL and MAL may cause interpretation errors. I believe that the effect of this is obvious in the plot of the MAL LR and PDR on 21-24 May.**

Yes you are right. So, we explain all this now in detail. But we leave the values in the figure, and discuss the findings on 20-23 May in detail to avoid possible confusion about the high depol values and lidar ratios in the MAL for the final 4 days of the cruise..

**OTHER POINTS:**

**26) Abstract P1 L13: add "at different longitude" after "20 nights". Exchange the order of the words MAL and SAL, as later on MAL is always presented before SAL. L17: add "typical" between "compared to" and "pure dust values".** Done!

**27) P2 L1-2: add "in very specific circumstances" before "be lifted up to the tropopause"**

We write: … under favorable meteorological conditions

**28) P4 L7 "the observed optical properties": it is unclear which observations the authors refer to: the current paper? pre-existing literature?**

We changed the text. We state more clearly that our measurements in Section 3 show this…..

**9) P8 L9-10: "are predicted or described by" --> "confirm"** Done!

**30) P8 L13: draw top of MBL in Fig. 2.**

We didn't do that. The clouds (white spots) nicely and precisely indicate the actual MBL top. Without the line, the reader has the chance to make his own opinion about the MBL top. And we have the MAL top in Figure 1 (top panel). That should be sufficient.

**31) P8 L15: "dark blue layers" --> "layers showing a low depolarization"** Done!

**32) P8 L24: "dissolve" --> "evaporate"** Done!

**33) P9 L1-3: are the NE and E winds in the MAL and SAL confirmed in the radiosondes launched aboard the ship?**

We checked our snapshot-like radiosonde profiles. We see a tendency from NE winds to E-winds. But this is not very pronounced. Only in case 2 the classical behavior is found. 60 deg wind direction in the MAL, and 100-120

deg in the SAL, wind speed is around 10 m/s in the MAL as well as in the SAL, with a peak of 14 m/s in the lower SAL (just above the trade wind inversion height). We state that in Section 3.2.1.

**34) P11 L25: add at end of paragraph: "However, the data presented here are insufficient to either support or deny such influence".** Done! (Sect. 3.3)

**35) P13 L15: "dust characteristics" --> "dust optical properties"** Done!

**36) P13 L25: add "according to the conceptual model" before "gravitational settling is responsible for the removal of dust"** Done!

**37) P13 L33: add "increasing" before "height"** Done!

**38) P14 L22: cite the Karyampudi paper again** Done!

**39) P15 L29: add "on board the research vessel" after "launched"** Done!

**40) Figure 1 readability: continents in top panel, and grey areas in bottom panel: they are not visible.**

We improved the map (in the top panel) as much as possible.

**41) Figure 2: draw MBL top height in the left panels; caption: "given in blue" --> "low depolarizations displayed in blue".**

As mentioned, we think to show the more important MAL top (or SAL base) is sufficient. We changed the caption text accordingly.

**42) Figure 6 readability: grey areas are not visible on printed copy.**

We did not make any changes. The printouts we made with the TROPOS institutes printer clearly show the gray areas… So, we do not see a reason to change the figure.

[revised manuscript text omitted]

---

## Author Response (AR3)

Dear Editor,

Thank you for careful reading and the suggestions.

We included your comments. The changes are given in bold in the paper (included below).

Abstract lines 5-6 - In order to provide the Kanitz reference and to fully distinguish what is presented in this new publication, please change to 'were reported by Kanitz et al., (2014). Here we present four observational cases representing key stages of the SAL evolution between Africa and the Caribbean in detail...'  **Done!**

abstract line 8 - 'extent'  **Improved!**

abstract line 20 - consider adding 'and back trajectories' after 'and depolarization ratio' since the back trajectories are an important factor in the identification of aerosol type here.  **Considered!**

Page 3 lines 22-25 - consider mentioning that you discuss the intermediate transport stages in addition to the two presented by Kanitz et al. In my opinion this is a useful and important addition of this paper. Yes, we agree. **We included such a statement.**

Page 4 line 12 - change to 'the SAL top subsides slowly'  **Done!**

Page 7 line 1 - 'results presented here differ only slightly' - this is not clear enough as it could refer to either quantitative data values, or more generally to results and conclusions drawn from the work. I assume the former is meant, so please change to 'quantitative data values presented here differ only slightly...' or similar. **We changed it accordingly.**

Page 8 line 4 - remove (?) **We left it in because it is a reference (wrongly written in TEX, therefore the questionmark appeared.)**

Page 11 line 7 - change to 'prove this argument' **Done!**

Page 17 line 28-29 - 'which would support that size-dependent gravitational settling has a strong impact on the dust amount in the SAL and significantly changes the dust size distribution with increasing transport time.' These words appear to contradict the main conclusions in the article that the size distribution/dust properties to not appear to change much with transport, and also appear to contradict the first part of the sentence. Please review/change/clarify this sentence. **Done!**

[revised manuscript text omitted]